# ACHIEVING MINIMAX OPTIMAL SAMPLE COMPLEXITY OF OFFLINE REINFORCEMENT LEARNING: A DRO-BASED APPROACH

## ABSTRACT

Offline reinforcement learning aims to learn from pre-collected datasets without active exploration. This problem faces significant challenges, including limited data availability and distributional shifts. Existing approaches adopt a pessimistic stance towards uncertainty by penalizing rewards of under-explored state-action pairs to estimate value functions conservatively. In this paper, we show that the distributionally robust optimization (DRO) based approach can also address these challenges and is minimax optimal. Specifically, we directly model the uncertainty in the transition kernel and construct an uncertainty set of statistically plausible transition kernels. We then find the policy that optimizes the worst-case performance over this uncertainty set. We first design a metric-based Hoeffding-style uncertainty set such that with high probability the true transition kernel is in this set. We prove that to achieve a sub-optimality gap of $\epsilon$, the sample complexity is $\mathcal{O}(SC^{\pi^*}\epsilon^{-2}(1-\gamma)^{-4})$, where $\gamma$ is the discount factor, $S$ is the number of states, and $C^{\pi^*}$ is the single-policy clipped concentrability coefficient which quantifies the distribution shift. To achieve the optimal sample complexity, we further propose a less conservative Bernstein-style uncertainty set, which, however, does not necessarily include the true transition kernel. We show that an improved sample complexity of $\mathcal{O}(SC^{\pi^*}\epsilon^{-2}(1-\gamma)^{-3})$ can be obtained, which matches with the minimax lower bound for offline reinforcement learning, and thus is minimax optimal.

## 1 INTRODUCTION

Reinforcement learning (RL) has achieved impressive empirical success in many domains, e.g., (Mnih et al., 2015; Silver et al., 2016). Nonetheless, most of the success stories rely on the premise that the agent can actively explore the environment and receive feedback to further promote policy improvement. This trial-and-error procedure can be costly, unsafe, or even prohibitory in many real-world applications, e.g., autonomous driving (Kiran et al., 2021) and health care (Yu et al., 2021a). To address the challenge, offline (or batch) reinforcement learning (Lange et al., 2012; Levine et al., 2020) was developed, which aims to learn a competing policy from a pre-collected dataset without access to online exploration.

A straightforward idea for offline RL is to use the pre-collected dataset to learn an estimated model of the environment, and then learn an optimal policy for this model. This approach performs well when the dataset sufficiently explored the environment, e.g., (Agarwal et al., 2020a). However, under more general offline settings, the static dataset can be limited, which results in the distribution shift challenge and inaccurate model estimation (Kidambi et al., 2020; Ross & Bagnell, 2012; Li et al., 2022). Namely, the pre-collected dataset is often restricted to a small subset of state-action pairs, and the behavior policy used to collect the dataset induces a state-action visitation distribution that is different from the one induced by the optimal policy. This distribution shift and the limited amount of data lead to uncertainty in the estimation of the model, i.e., transition kernel and/or reward function.

To address the above challenge, one natural approach is to first quantify the uncertainty, and further take a pessimistic (conservative) approach in face of such uncertainty. Despite of the fact that the uncertainty exists in the transition kernel estimate, existing studies mostly take the approach to

penalize the reward function for less-visited state-action pairs to obtain a pessimistic estimation of the value function, known as the Lower Confidence Bound (LCB) approach (Rashidinejad et al., 2021; Li et al., 2022; Shi et al., 2022; Yan et al., 2022). In this paper, we develop a direct approach to analyzing such uncertainty in the transition kernel by constructing a statistically plausible set of transition kernels, i.e., uncertainty set, and optimizing the worst-case performance over this uncertainty set. This principle is referred to as "distributionally robust optimization (DRO)" in the literature (Nilim & El Ghaoui, 2004; Iyengar, 2005). This DRO-based approach directly tackles the uncertainty in the transition kernel. We show that our approach achieves the minimax optimal sample complexity (Rashidinejad et al., 2021). We summarize our major contributions as follows.

## 1.1 MAIN CONTRIBUTIONS

In this work, we focus on the most general partial coverage setting (see Section 2.3 for the definition). We develop a DRO-based framework that efficiently solves the offline RL problem. More importantly, we design a Bernstein-style uncertainty set and show that its sample complexity is minimax optimal.

**DRO-based Approach Solves Offline RL.** We construct a Hoeffding-style uncertainty set centered at the empirical transition kernel to guarantee that with high probability, the true transition kernel lies within the uncertainty set. Then, optimizing the worst-case performance over the uncertainty set provides a lower bound on the performance under the true environment. Our uncertainty model enables easy solutions using the robust dynamic programming approach developed for robust MDP in (Iyengar, 2005; Nilim & El Ghaoui, 2004) within a polynomial computational complexity. We further show the sample complexity to achieve an $\epsilon$-optimal policy using our approach is $\mathcal{O}\left(\frac{SC^{\pi^*}}{(1-\gamma)^4\epsilon^2}\right)$ (up to a log factor), where $\gamma$ is the discount factor, and $C^{\pi^*}$ is the single-policy concentrability for any comparator policy $\pi^*$ (see Definition 1). This sample complexity matches with the best-known complexity of the LCB and Hoeffding-style method (Rashidinejad et al., 2021; Yan et al., 2022), which demonstrates that our DRO framework can directly tackle the model uncertainty and effectively solve offline RL.

**Achieving Minimax Optimality via Design of Bernstein-style Uncertainty Set.** While the approach described above is effective in achieving an $\epsilon$-optimal policy with relatively low sample complexity, it tends to exhibit excessive conservatism as its complexity surpasses the minimax lower bound for offline RL algorithms (Rashidinejad et al., 2021) by a factor of $(1-\gamma)^{-1}$. To close this gap, we discover that demanding the true transition kernel to be within the uncertainty set with high probability, i.e., the true environment and the worst-case one are close, can be overly pessimistic and unnecessary. What is of paramount importance is that the value function under the worst-case transition kernel within the uncertainty set (almost) lower bounds the one under the true transition kernel. Notably, this requirement is considerably less stringent than mandating that the actual kernel be encompassed by the uncertainty set. We then design a less conservative Bernstein-style uncertainty set, which has a smaller radius and thus is a subset of the Hoeffding-style uncertainty set. We prove that to obtain an $\epsilon$-optimal policy, the sample complexity is $\mathcal{O}\left(\frac{SC^{\pi^*}}{(1-\gamma)^3\epsilon^2}\right)$. This complexity indicates the minimax optimality of our approach by matching with the minimax lower bound of the sample complexity for offline RL (Rashidinejad et al., 2021) and the best result from the LCB approach (Li et al., 2022).

## 1.2 RELATED WORKS

There has been a proliferation of works on offline RL. In this section, we mainly discuss works on model-based approaches. There are also model-free approaches, e.g., (Liu et al., 2020; Kumar et al., 2020; Agarwal et al., 2020b; Yan et al., 2022; Shi et al., 2022), which are not the focus here.

**Offline RL under global coverage.** Existing studies on offline RL often make assumptions on the coverage of the dataset. This can be measured by the distribution shift between the behavior policy and the occupancy measure induced by the target policy, which is referred to as the concentrability coefficient (Munos, 2007; Rashidinejad et al., 2021). Many previous works, e.g., (Scherrer, 2014; Chen & Jiang, 2019; Jiang, 2019; Wang et al., 2019; Liao et al., 2020; Liu et al., 2019; Zhang et al., 2020a; Munos & Szepesvari, 2008; Uehara et al., 2020; Duan et al., 2020; Xie & Jiang, 2020; Levine et al., 2020; Antos et al., 2007; Farahmand et al., 2010), assume that the density ratio between the

above two distributions is finite for all state-action pairs and policies, which is known as global coverage. This assumption essentially requires the behavior policy to be able to visit all possible state-action pairs, which is often violated in practice (Gulcehre et al., 2020; Agarwal et al., 2020b; Fu et al., 2020).

**Offline RL under partial coverage.** Recent studies relax the above assumption of global coverage to partial coverage or single-policy concentrability. Partial coverage assumes that the density ratio between the distributions induced by a single target policy and the behavior policy is bounded for all state-action pairs. Therefore, this assumption does not require the behavior policy to be able to visit all possible state-action pairs, as long as it can visit those state-actions pairs that the target policy will visit. This partial coverage assumption is more feasible and applicable in real-world scenarios. In this paper, we focus on this practical partial coverage setting. Existing approaches under the partial coverage assumption can be divided into three categories as follows.

*Regularized Policy Search.* The first approach regularizes the policy so that the learned policy is close to the behavior policy (Fujimoto et al., 2019b; Wu et al., 2019; Jaques et al., 2019; Peng et al., 2019; Siegel et al., 2020; Wang et al., 2020; Kumar et al., 2019; Fujimoto et al., 2019a; Ghasemipour et al., 2020; Nachum et al., 2019; Zhang et al., 2020b; 2023). Thus, the learned policy is similar to the behavior policy which generates the dataset, hence this approach works well when the dataset is collected from experts (Wu et al., 2019; Fu et al., 2020).

*Reward Penalization or LCB Approaches.* One of the most widely used approaches is to penalize the reward in face of uncertainty to obtain an pessimistic estimation that lower bounds the real value function, e.g., (Kidambi et al., 2020; Yu et al., 2020; 2021b; Buckman et al., 2020; Jin et al., 2021; Xie et al., 2021; Yin & Wang, 2021; Liu et al., 2020; Cui & Du, 2022; Chen et al., 2021; Zhong et al., 2022). The most popular and potential approach VI-LCB (Rashidinejad et al., 2021; Li et al., 2022) penalizes the reward with a bonus term that is inversely proportional to the number of samples. The tightest sample complexity is obtained in (Li et al., 2022) by designing a Bernstein-style penalty term, which matches the minimax lower bound in (Rashidinejad et al., 2021).

*DRO-based Approaches.* Another approach is to first construct a set of "statistically plausible" MDP models based on the empirical transition kernel, and then find the policy that optimizes the worst-case performance over this set (Zanette et al., 2021; Uehara & Sun, 2021; Rigter et al., 2022; Bhardwaj et al., 2023; Guo et al., 2022; Hong et al., 2023; Chang et al., 2021; Blanchet et al., 2023). However, finding such a policy under the models proposed in these works can be NP-hard, hence some heuristic approximations without theoretical optimality guarantee are used to deploy their approaches. Our work falls into this category, but the computational complexity is polynomial, and its sample complexity is minimax optimal.

**Robust RL with distributional uncertainty.** In this paper, our algorithm is based on the framework of robust MDP (Iyengar, 2005; Nilim & El Ghaoui, 2004; Bagnell et al., 2001; Satia & Lave Jr, 1973; Wiesemann et al., 2013), which finds the policy with the best worst-case performance over an uncertainty set of transition dynamics. When the uncertainty set is fully known, the problem can be solved by robust dynamic programming. The sample complexity of model-based approaches without full knowledge of the uncertainty sets were studied in, e.g., (Yang et al., 2021; Xu et al., 2023; Panaganti & Kalathil, 2022; Shi et al., 2023; Panaganti & Kalathil, 2022), where a generative model is typically assumed. This model-based approach is further adapted to the robust offline setting in (Panaganti et al., 2022; Shi & Chi, 2022). Yet in these works, the challenge of partial coverage is addressed using the LCB aproach, i.e., penalizing the reward functions, whereas we show that the DRO framework itself can also address the challenge of partial coverage in the offline setting.

## 2 PRELIMINARIES

### 2.1 MARKOV DECISION PROCESS (MDP)

An MDP can be characterized by a tuple $(\mathcal{S}, \mathcal{A}, \mathsf{P}, r)$, where $\mathcal{S}$ and $\mathcal{A}$ are the state and action spaces, $\mathsf{P} = \{\mathsf{P}_s^a \in \Delta(\mathcal{S}), a \in \mathcal{A}, s \in \mathcal{S}\}$[1] is the transition kernel, $r : \mathcal{S} \times \mathcal{A} \to [0, 1]$ is the deterministic reward function, and $\gamma \in [0, 1)$ is the discount factor. Specifically, $\mathsf{P}_s^a = (p_{s,s'}^a)_{s' \in \mathcal{S}}$, where $p_{s,s'}^a$ denotes the probability that the environment transits to state $s'$ if taking action $a$ at state $s$. The

---

[1]$\Delta(\mathcal{S})$ denotes the probability simplex defined on $\mathcal{S}$.

reward of taking action $a$ at state $s$ is given by $r(s, a)$. A stationary policy $\pi$ is a mapping from $\mathcal{S}$ to a distribution over $\mathcal{A}$, which indicates the probabilities of the agent taking actions at each state. At each time $t$, an agent takes an action $a_t \sim \pi(s_t)$ at state $s_t$, the environment then transits to the next state $s_{t+1}$ with probability $p^{a_t}_{s_t, s_{t+1}}$, and the agent receives reward $r(s_t, a_t)$.

The value function of a policy $\pi$ starting from any initial state $s \in \mathcal{S}$ is defined as the expected accumulated discounted reward by following $\pi$: $V^{\pi}_{\mathsf{P}}(s) \triangleq \mathbb{E}_{\mathsf{P}} \left[ \sum_{t=0}^{\infty} \gamma^t r(S_t, A_t) | S_0 = s, \pi \right]$, where $\mathbb{E}_{\mathsf{P}}$ denotes the expectation when the state transits according to $\mathsf{P}$. Let $\rho$ denote the initial state distribution, and denote the value function under the initial distribution $\rho$ by $V^{\pi}_{\mathsf{P}}(\rho) \triangleq \mathbb{E}_{s \sim \rho}[V^{\pi}_{\mathsf{P}}(s)]$.

## 2.2 ROBUST MARKOV DECISION PROCESS

In the robust MDP, the transition kernel is not fixed and lies in some uncertainty set $\mathcal{P}$. Define the robust value function of a policy $\pi$ as the worst-case expected accumulated discounted reward over the uncertainty set: $V^{\pi}_{\mathcal{P}}(s) \triangleq \min_{\mathsf{P} \in \mathcal{P}} \mathbb{E}_{\mathsf{P}} \left[ \sum_{t=0}^{\infty} \gamma^t r(S_t, A_t) | S_0 = s, \pi \right]$. Similarly, the robust action-value function for a policy $\pi$ is defined as $Q^{\pi}_{\mathcal{P}}(s, a) = \min_{\mathsf{P} \in \mathcal{P}} \mathbb{E}_{\mathsf{P}} \left[ \sum_{t=0}^{\infty} \gamma^t r(S_t, A_t) | S_0 = s, A_0 = a, \pi \right]$. The goal of robust RL is to find the optimal robust policy that maximizes the worst-case accumulated discounted reward, i.e., $\pi_r = \arg\max_{\pi} V^{\pi}_{\mathcal{P}}(s), \forall s \in \mathcal{S}$. It is shown in (Iyengar, 2005; Nilim & El Ghaoui, 2004; Wiesemann et al., 2013) that the optimal robust value function is the unique solution to the optimal robust Bellman equation $V^{\pi_r}_{\mathcal{P}}(s) = \max_a \{r(s, a) + \gamma \sigma_{\mathcal{P}^a_s}(V^{\pi_r}_{\mathcal{P}})\}$, where $\sigma_{\mathcal{P}^a_s}(V) \triangleq \min_{p \in \mathcal{P}^a_s} p^\top V$ denotes the support function of $V$ on a set $\mathcal{P}^a_s$ and the corresponding robust Bellman operator is a $\gamma$-contraction.

## 2.3 OFFLINE REINFORCEMENT LEARNING

Under the offline setting, the agent cannot interact with the MDP and instead is given a pre-collected dataset $\mathcal{D}$ consisting of $N$ tuples $\{(s_i, a_i, s'_i, r_i) : i = 1, ..., N\}$, where $r_i = r(s_i, a_i)$ is the deterministic reward, and $s'_i \sim \mathsf{P}^{a_i}_{s_i}$ follows the transition kernel $\mathsf{P}$ of the MDP. The $(s_i, a_i)$ pairs in $\mathcal{D}$ are generated i.i.d. according to an unknown data distribution $\mu$ over the state-action space. In this paper, we consider the setting where the reward functions $r$ is deterministic but unknown. We denote the number of samples transitions from $(s, a)$ in $\mathcal{D}$ by $N(s, a)$, i.e., $N(s, a) = \sum_{i=1}^{N} \mathbf{1}_{(s_i, a_i) = (s, a)}$ and $\mathbf{1}_{X=x}$ is the indicator function.

The goal of offline RL is to find a policy $\pi$ which optimizes the value function $V^{\pi}_{\mathsf{P}}$ based on the offline dataset $\mathcal{D}$. Let $d^{\pi}$ denote the discounted occupancy distribution associated with $\pi$: $d^{\pi}(s) = (1 - \gamma) \sum_{t=0}^{\infty} \gamma^t \mathbb{P}(S_t = s | S_0 \sim \rho, \pi, \mathsf{P})$. In this paper, we focus on the partial coverage setting and adopt the following definition from (Li et al., 2022) to measure the distribution shift between the dataset distribution and the occupancy measure induced by a single policy $\pi^*$:

**Definition 1.** *(Single-policy clipped concentrability) The single-policy clipped concentrability coefficient of a policy $\pi^*$ is defined as*

$$C^{\pi^*} \triangleq \max_{s,a} \frac{\min\{d^{\pi^*}(s, a), \frac{1}{S}\}}{\mu(s, a)}, \tag{1}$$

*where $S \triangleq |\mathcal{S}|$ denotes the number of states.*

We note another unclipped version of $C^{\pi^*}$ is also commonly used in the literature, e.g., (Rashidinejad et al., 2021; Uehara & Sun, 2021), defined as $\tilde{C}^{\pi^*} \triangleq \max_{s,a} \frac{d^{\pi^*}(s,a)}{\mu(s,a)}$. It is straightforward to verify that $C^{\pi^*} \leq \tilde{C}^{\pi^*}$, and all of our results remain valid if $C^{\pi^*}$ is replaced by $\tilde{C}^{\pi^*}$. A more detailed discussion can be found in (Li et al., 2022; Shi & Chi, 2022).

The goal of this paper is to find a policy $\pi$ which minimizes the sub-optimality gap compared to a comparator policy $\pi^*$ under some initial state distribution $\rho$: $V^{\pi^*}_{\mathsf{P}}(\rho) - V^{\pi}_{\mathsf{P}}(\rho)$.

## 3 OFFLINE RL VIA DISTRIBUTIONALLY ROBUST OPTIMIZATION

Model-based methods usually commence by estimating the transition kernel employing its maximum likelihood estimate. Nevertheless, owing to the inherent challenges associated with distribution shift

and limited data in the offline setting, uncertainties can arise in these estimations. For instance, the dataset may not encompass every state-action pair, and the sample size may be insufficient to yield a precise estimate of the transition kernel. In this paper, we directly quantify the uncertainty in the empirical estimation of the transition kernel, and construct a set of "statistically possible" transition kernels, referred to as uncertainty set, so that it encompasses the actual environment. We then employ the DRO approach to optimize the worst-case performance over the uncertainty set. Notably, this formulation essentially transforms the problem into a robust Markov Decision Process (MDP), as discussed in Section 2.2.

In Section 3.1, we first introduce a direct metric-based Hoeffding-style approach to construct the uncertainty set such that the true transition kernel is in the uncertainty set with high probability. We then present the robust value iteration algorithm to solve the DRO problem. We further theoretically characterize the bound on the sub-optimality gap and show that the sample complexity to achieve an $\epsilon$-optimality gap is $\mathcal{O}((1-\gamma)^{-4}\epsilon^{-2}SC^{\pi^*})$. This gap matches with the best-known sample complexity for the LCB method using the Hoeffding-style bonus term (Rashidinejad et al., 2021). This result shows the effectiveness of our DRO-based approach in solving the offline RL problem.

We then design a less conservative Bernstein-style uncertainty set aiming to achieve the minimax optimal sample complexity in Section 3.2. We theoretically establish that our approach attains an enhanced and minimax optimal sample complexity of $\mathcal{O}\left((1-\gamma)^{-3}\epsilon^{-2}SC^{\pi^*}\right)$. Notably, this sample complexity matches with the minimax lower bound (Rashidinejad et al., 2021) and stands on par with the best results achieved using the LCB approach (Li et al., 2022).

Our approach starts with learning an empirical model of the transition kernel and reward from the dataset as follows. These empirical transition kernels will be used as the centroid of the uncertainty set. For any $(s, a)$, if $N(s, a) > 0$, set

$$\hat{\mathsf{P}}_{s,s'}^a = \frac{\sum_{i \leq N} \mathbf{1}_{(s_i,a_i,s_i')=(s,a,s')}}{N(s,a)}, \hat{r}(s,a) = r_i(s,a); \tag{2}$$

And if $N(s, a) = 0$, set

$$\hat{\mathsf{P}}_{s,s'}^a = \mathbf{1}_{s'=s}, \hat{r}(s,a) = 0. \tag{3}$$

The MDP $\hat{\mathsf{M}} = (\mathcal{S}, \mathcal{A}, \hat{\mathsf{P}}, \hat{r})$ with the empirical transition kernel $\hat{\mathsf{P}}$ and empirical reward $\hat{r}$ is referred to as the empirical MDP. For unseen state-action pairs $(s, a)$ in the offline dataset, we take a conservative approach and let the estimated $\hat{r}(s,a) = 0$, and set $s$ as an absorbing state if taking action $a$. Then the action value function at $(s, a)$ for the empirical MDP shall be zero, which discourages the choice of action $a$ at state $s$.

For each state-action pair $(s, a)$, we construct an uncertainty set centered at the empirical transition kernel $\hat{\mathsf{P}}_s^a$, with a radius inversely proportional to the number of samples in the dataset. Specifically, set the uncertainty set as $\hat{\mathcal{P}} = \bigotimes_{s,a} \hat{\mathcal{P}}_s^a$ and

$$\hat{\mathcal{P}}_s^a = \left\{ q \in \Delta(\mathcal{S}) : D(q, \hat{\mathsf{P}}_s^a) \leq R_s^a \right\}, \tag{4}$$

where $D(\cdot, \cdot)$ is some function that measures the difference between two probability distributions, e.g., total variation, Chi-square divergence, $R_s^a$ is the radius ensuring that the uncertainty set adapts to the dataset size and the degree of confidence, which will be determined later. We then construct the robust MDP as $\hat{\mathcal{M}} = (\mathcal{S}, \mathcal{A}, \hat{\mathcal{P}}, \hat{r})$.

As we shall show later, the optimal robust policy

$$\pi_r = \arg\max_\pi \min_{\mathsf{P} \in \hat{\mathcal{P}}} V_\mathsf{P}^\pi(s), \forall s \in \mathcal{S}. \tag{5}$$

w.r.t. $\hat{\mathcal{M}}$ performs well in the real environment and reaches a small sub-optimality gap. In our construction in eq. (4), the uncertainty set is $(s, a)$-rectangular, i.e., for different state-action pairs, the corresponding uncertainty sets are independent. With this rectangular structure, the optimal robust policy can be found by utilizing the robust value iteration algorithm or robust dynamic programming (Algorithm 1), and the corresponding robust value iteration at each step can be solved in polynomial time (Wiesemann et al., 2013). In contrast, the uncertainty set constructed in (Uehara

& Sun, 2021; Bhardwaj et al., 2023), defined as $\mathcal{T} = \{Q : \mathbb{E}_{\mathcal{D}}[\|\hat{P}_s^a - Q_s^a\|^2] \leq \zeta\}$, does not enjoy such a rectangularity. Solving a robust MDP with such an uncertainty set can be, however, NP-hard (Wiesemann et al., 2013). The approach developed in (Rigter et al., 2022) to solve it is based on the heuristic approach of adversarial training, and therefore is lack of theoretical guarantee.

---

**Algorithm 1** Robust Value Iteration (Nilim & El Ghaoui, 2004; Iyengar, 2005)

---

**INPUT**: $\hat{r}, \hat{\mathcal{P}}, V, \mathcal{D}$
 1: **while** TRUE **do**
 2:  **for** $s \in \mathcal{S}$ **do**
 3:   $V(s) \leftarrow \max_a \{\hat{r}(s, a) + \gamma \sigma_{\hat{\mathcal{P}}_s^a}(V)\}$
 4:  **end for**
 5: **end while**
 6: **for** $s \in \mathcal{S}$ **do**
 7:  $\pi_r(s) \in \arg\max_{a \in \mathcal{A}}\{\hat{r}(s, a) + \gamma \sigma_{\hat{\mathcal{P}}_s^a}(V)\}\}$
 8: **end for**
**Output**: $\pi_r$

---

The algorithm converges to the optimal robust policy linearly since the robust Bellman operator is a $\gamma$-contraction (Iyengar, 2005). The computational complexity of the support function $\sigma_{\hat{\mathcal{P}}_s^a}(V)$ in Lines 3 and 7 w.r.t. the uncertainty sets we constructed matches the ones of the LCB approaches (Rashidinejad et al., 2021; Li et al., 2022).

In the following two sections, we specify the constructions of the uncertainty sets.

### 3.1 HOEFFDING-STYLE RADIUS

We first employ the total variation to construct this uncertainty set. Specifically, we let $D$ be the total variation distance and $R_s^a \triangleq \min\left\{1, \sqrt{\frac{\log \frac{SA}{\delta}}{8N(s,a)}}\right\}$. The radius is inversely proportional to the number of samples. Fewer samples result in a larger uncertainty set and imply that we should be more conservative in estimating the transition dynamics at this state-action pair. Other distance function of $D$ can also be used, contingent upon the concentration inequality being applied.

In Algorithm 1, $\sigma_{\hat{\mathcal{P}}_s^a}(V) = \min_{q \in \hat{\mathcal{P}}_s^a}\{q^\top V\}$ can be equivalently solved by solving its dual form (Iyengar, 2005), which is a convex optimization problem: $\max_{0 \leq \mu \leq V}\{\hat{P}_s^a(V - \mu) - R_s^a \text{Span}(V - \mu)\}$, and $\text{Span}(X) = \max_i X(i) - \min_i X(i)$ is the span semi-norm of vector $X$. The computational complexity associated with solving it is $\mathcal{O}(S\log(S))$. Notably, this polynomial computational complexity is on par with the complexity of the VI-LCB approach (Li et al., 2022).

We then show that with this Hoeffding-style radius, the true transition kernel falls into the uncertainty set with high probability.

**Lemma 1.** *With probability at least $1 - \delta$, it holds that for any $s, a$, $P_s^a \in \hat{\mathcal{P}}_s^a$, i.e., $\|P_s^a - \hat{P}_s^a\| \leq R_s^a$.*

This result implies that the real environment $P$ falls into the uncertainty set $\hat{\mathcal{P}}$ with high probability, and hence finding the optimal robust policy of $\hat{\mathcal{M}}$ provides a worst-case performance guarantee. We further present our result of the sub-optimality gap in the following theorem.

**Theorem 1.** *Consider an arbitrary deterministic comparator policy $\pi^*$. With probability at least $1 - 2\delta$, the output policy $\pi_r$ of Algorithm 1 satisfies*

$$V_P^{\pi^*}(\rho) - V_P^{\pi_r}(\rho) \leq \frac{16SC^{\pi^*}\log\frac{NS}{\delta}}{(1 - \gamma)^2 N} + \sqrt{\frac{96SC^{\pi^*}\log\frac{SA}{\delta}}{(1 - \gamma)^4 N}}. \tag{6}$$

To achieve an $\epsilon$-optimality gap, a dataset of size $N = \mathcal{O}\left((1 - \gamma)^{-4}\epsilon^{-2}SC^{\pi^*}\right)$ is required for a Hoeffding-style uncertainty model. This sample complexity matches with the best-known sample complexity for LCB methods with Hoeffding-style bonus term (Rashidinejad et al., 2021). It suggests that our DRO-based approach can effectively address the offline RL problem.

However, there is still a gap between this sample complexity and the minimax lower bound in (Rashidinejad et al., 2021) and the best-known sample complexity of LCB-based method (Li et al., 2022), which is $\mathcal{O}\left((1-\gamma)^{-3}\epsilon^{-2}SC^{\pi^*}\right)$. We will address this problem via a Bernstein-style uncertainty set design in the next subsection.

## 3.2 BERNSTEIN-STYLE RADIUS

As discussed above, using a Hoeffding-style radius is able to achieve an $\epsilon$-optimal policy, however, with an unnecessarily large sample complexity. Compared with the minimax lower bound and the tightest result obtained in (Li et al., 2022), there exists a gap of order $\mathcal{O}((1-\gamma)^{-1})$. This gap is mainly because the Hoeffding-style radius is overly conservative. The idea of Hoeffding-style radius can be viewed as *distribution based*. That is, to construct the uncertainty set $\hat{\mathcal{P}}$ centered at $\hat{\mathsf{P}}$ large enough such that the true transition kernel $\mathsf{P}$ falls into $\hat{\mathcal{P}}$ with high probability (Lemma 1). Therefore, it holds that $V_{\hat{\mathcal{P}}}^{\pi_r} \leq V_{\mathsf{P}}^{\pi_r}$ and the sub-optimality gap can be bounded as

$$V_{\mathsf{P}}^{\pi^*} - V_{\mathsf{P}}^{\pi_r} = \underbrace{V_{\mathsf{P}}^{\pi^*} - V_{\hat{\mathcal{P}}}^{\pi_r}}_{\Delta_1} + \underbrace{V_{\hat{\mathcal{P}}}^{\pi_r} - V_{\mathsf{P}}^{\pi_r}}_{\Delta_2} \leq \Delta_1. \tag{7}$$

However, $\Delta_2 = \min_{q \in \hat{\mathcal{P}}} \mathbb{E}_q[\sum \gamma^t r_t] - \mathbb{E}_{\mathsf{P}}[\sum \gamma^t r_t]$ is in fact the difference between the expectations under two different distributions, which shall be bounded tighter than merely using the distance of the two distributions. Hence we can instead construct a smaller uncertainty set, which may not cover the true transition kernel but can still imply a tight error bound of $\Delta_2$.

Specifically, note that $\Delta_2 = \underbrace{V_{\hat{\mathcal{P}}}^{\pi_r} - V_{\hat{\mathsf{P}}}^{\pi_r}}_{(a)} + \underbrace{V_{\hat{\mathsf{P}}}^{\pi_r} - V_{\mathsf{P}}^{\pi_r}}_{(b)}$, the idea hence is to choose a radius such that the sum of term $(a)$, which is the difference between the performance under the centroid kernel $\hat{\mathsf{P}}$ and the worst-case performance, and term $(b)$ to be small. We then construct the Bernstein-style uncertainty set as follows. Instead of total variation, we construct the uncertainty set using the Chi-square divergence, i.e., $D(p,q) = \chi^2(p||q) = \sum_s q(s)\left(1 - \frac{p(s)}{q(s)}\right)^2$. The reason why we adapt the Chi-square divergence instead of the total variation will be discussed later. We further set the radius as $R_s^a \triangleq \frac{48 \log \frac{4N}{\delta}}{N(s,a)}$, and construct the robust MDP $\hat{\mathcal{M}} = (\mathcal{S}, \mathcal{A}, \hat{\mathcal{P}} = \bigotimes_{s,a} \hat{\mathcal{P}}_s^a, \hat{r})$.

**Remark 1.** *From Pinsker's inequality and the fact that $D_{KL}(p||q) \leq \chi^2(p||q)$ (Nishiyama & Sason, 2020), it holds that $\|p - q\| \leq \sqrt{2\chi^2(p||q)}$. Hence the Bernstein-style uncertainty set is a subset of the Hoeffding-style uncertainty set in Section 3.1, and is less conservative.*

Similarly, we find the optimal robust policy w.r.t. the corresponding robust MDP $\hat{\mathcal{M}} = (\mathcal{S}, \mathcal{A}, \hat{\mathcal{P}}, \hat{r})$ using the robust value iteration with a slight modification, which is presented in Algorithm 2. Specifically, the output policy $\pi_r$ in Algorithm 2 is set to be the greedy policy satisfying $N(s,a) > 0$ if $N(s) > 0$. The existence of such a policy is proved in Lemma 3 in the appendix. This is to guarantee that when there is a tie of taking greedy actions, we will take an action that has appeared in the pre-collected dataset $\mathcal{D}$.

The support function $\sigma_{\hat{\mathcal{P}}_s^a}(V)$ w.r.t. the Chi-square divergence uncertainty set can also be computed using its dual form (Iyengar, 2005): $\sigma_{\hat{\mathcal{P}}_s^a}(V) = \max_{\alpha \in [V_{\min}, V_{\max}]}\{\hat{\mathsf{P}}_s^a V_\alpha - \sqrt{R_s^a \mathbf{Var}_{\hat{\mathsf{P}}_s^a}(V_\alpha)}\}$, where $V_\alpha(s) = \min\{\alpha, V(s)\}$. The dual form is also a convex optimization problem and can be solved efficiently within a polynomial time $\mathcal{O}(S \log S)$ (Iyengar, 2005).

Using the Chi-square divergence enables a smaller radius and yields a tighter bound on $\Delta_2 = (a)+(b)$. Namely, $(b)$ can be bounded by a $N^{-0.5}$-order bound according to the Bernstein's inequality (see Lemma 6 in the Appendix). Simultaneously, our goal is to obtain a bound with the same order on $(a)$, which effectively offsets the bound on $(b)$, and yields a tighter bound on $\Delta_2$. The robust value function w.r.t. the total variation uncertainty set, however, depends on $R_s^a$ linearly (see the dual form we discussed above); On the other hand, the solution to the Chi-square divergence uncertainty set incorporates a term of $\sqrt{R_s^a}$ which enables us to set a lower-order radius (i.e., set $R_s^a = (N(s,a))^{-1}$) to offset the $N^{-0.5}$-order bound on $(b)$.

---

**Algorithm 2** Robust Value Iteration

---

**INPUT**: $\hat{r}, \hat{\mathcal{P}}, V, \mathcal{D}$
1: **while** TRUE **do**
2:    **for** $s \in \mathcal{S}$ **do**
3:       $N(s) \leftarrow \sum_{i=1}^{N} \mathbf{1}_{(s_i)=s}$
4:       $V(s) \leftarrow \max_a\{\hat{r}(s,a) + \gamma\sigma_{\hat{\mathcal{P}}_s^a}(V)\}$
5:    **end for**
6: **end while**
7: **for** $s \in \mathcal{S}$ **do**
8:    **if** $N(s) > 0$ **then**
9:       $\pi_r(s) \in \{\arg\max_{a \in \mathcal{A}}\{\hat{r}(s,a) + \gamma\sigma_{\hat{\mathcal{P}}_s^a}(V)\}\} \cap \{a : N(s,a) > 0\}$
10:   **end if**
11:   $\pi_r(s) \in \arg\max_{a \in \mathcal{A}}\{\hat{r}(s,a) + \gamma\sigma_{\hat{\mathcal{P}}_s^a}(V)\}\}$
12: **end for**
**Output**: $\pi_r$

---

We then characterize the optimality gap obtained from Algorithm 2 in the following theorem.

**Theorem 2.** *If $N \geq \frac{1}{(1-\gamma)\mu_{\min}^2}$, then the output policy $\pi_r$ of Algorithm 2 satisfies*

$$V_{\mathsf{P}}^{\pi^*}(\rho) - V_{\mathsf{P}}^{\pi_r}(\rho) \leq \sqrt{\frac{KSC^{\pi^*}\log\frac{4N}{\delta}}{(1-\gamma)^3 N}}, \tag{8}$$

*with probability at least $1 - 4\delta$, where $\mu_{\min} = \min\{\mu(s,a) : \mu(s,a) > 0\}$ denotes the minimal non-zero probability of $\mu$.*

Theorem 2 implies that our DRO approach can achieve an $\epsilon$-optimality gap, as long as the size of the dataset exceeds the order of

$$\mathcal{O}\left(\underbrace{\frac{SC^{\pi^*}}{(1-\gamma)^3\epsilon^2}}_{\epsilon\text{-dependent}} + \underbrace{\frac{1}{(1-\gamma)\mu_{\min}^2}}_{\text{burn-in cost}}\right). \tag{9}$$

The burn-in cost term indicates that the asymptotic bound of the sample complexity becomes relevant after the dataset size surpasses the burn-in cost. It represents the minimal requirement for the amount of data. In fact, if the dataset is too small, we should not expect to learn a well-performed policy from it. Burn-in cost also widely exists in the sample complexity studies of RL, e.g., $H^9 SC^{\pi^*}$ in (Xie et al., 2021), $\frac{S^3 A^2}{(1-\gamma)^4}$ in (He et al., 2021), and $\frac{SC^{\pi^*}}{(1-\gamma)^5}$ in (Yan et al., 2022), $\frac{H}{\mu_{\min}p_{\min}}$ in (Shi & Chi, 2022). Note that the burn-in cost term is independent of the accuracy level $\epsilon$, which implies the sample complexity is less than $\mathcal{O}\left(\frac{SC^{\pi^*}}{(1-\gamma)^3\epsilon^2}\right)$, as long as $\epsilon$ is small. This result obtains the optimal complexity according to the minimax lower bound in (Rashidinejad et al., 2021), and also matches the tightest bound obtained using the LCB approach (Li et al., 2022). This suggests that our DRO approach can effectively address offline RL while imposing minimal demands on the dataset, thus optimizing the sample complexity associated with offline RL.

We compare our results with the most related works in Table 1. Our approach is the first model-uncertainty-based approach obtaining the minimax optimal sample complexity in offline RL.

## 4 EXPERIMENTS

We adapt our DRO framework under two problems, the Garnet problem $\mathcal{G}(30, 20)$ (Archibald et al., 1995), and the Frozen-Lake problem (Brockman et al., 2016) to numerically verify our results.

In the Garnet problem, $|\mathcal{S}| = 30$ and $|\mathcal{A}| = 20$. The transition kernel $\mathsf{P} = \{\mathsf{P}_s^a, s \in \mathcal{S}, a \in \mathcal{A}\}$ is randomly generated following a normal distribution: $\mathsf{P}_s^a \sim \mathcal{N}(\omega_s^a, \sigma_s^a)$ and then normalized, and the reward function $r(s,a) \sim \mathcal{N}(\nu_s^a, \psi_s^a)$, where $\omega_s^a, \sigma_s^a, \nu_s^a, \psi_s^a \sim \mathbf{Uniform}[0, 100]$.

| | Approach Type | Sample Complexity | Computational Cost |
|---|---|---|---|
| Our Approach | DRO | $\mathcal{O}\left(\frac{SC^{\pi^*}}{\epsilon^2(1-\gamma)^3}\right)$ | Polynomial |
| (Rashidinejad et al., 2021) | LCB | $\mathcal{O}\left(\frac{SC^{\pi^*}}{\epsilon^2(1-\gamma)^5}\right)$ | Polynomial |
| (Uehara & Sun, 2021) | DRO | $\mathcal{O}\left(\frac{SC^{\pi^*}}{\epsilon^2(1-\gamma)^4}\right)$ | NP-Hard |
| (Li et al., 2022) | LCB | $\mathcal{O}\left(\frac{SC^{\pi^*}}{\epsilon^2(1-\gamma)^3}\right)$ | Polynomial |
| (Rashidinejad et al., 2021) | Minimax Lower bound | $\mathcal{O}\left(\frac{SC^{\pi^*}}{\epsilon^2(1-\gamma)^3}\right)$ | - |

Table 1: Comparision with related works.

In the Frozen-Lake problem, an agent aim to cross a $4 \times 4$ frozen lake from Start to Goal without falling into any Holes by walking over the frozen lake.

In both problems, we deploy our approach under both global coverage and partial coverage conditions. Specifically, under the global coverage setting, the dataset is generated by the uniform policy $\pi(a|s) = \frac{1}{|\mathcal{A}|}$; And under the partial coverage condition, the dataset is generated according to $\mu(s,a) = \frac{\mathbf{1}_{a=\pi^*(s)}}{2} + \frac{\mathbf{1}_{a=\eta}}{2}$, where $\eta$ is an action randomly chosen from the action space $\mathcal{A}$.

At each time step, we generate 40 new samples and add them to the offline dataset and deploy our DRO approach on it. We also deploy the LCB approach (Li et al., 2022) and non-robust model-based dynamic programming as the baselines. We run the algorithms independently 10 times and plot the average value of the sub-optimality gaps over all 10 trajectories. We also plot the 95th and 5th percentiles of the 10 curves as the upper and lower envelopes of the curves. The results are presented in Figure 1. It can be seen from the results that our DRO approach finds the optimal policy with relatively less data; The LCB approach has a similar convergence rate to the optimal policy, which verifies our theoretical results; The non-robust DP converges much slower, and can even converge to a sub-optimal policy. The results hence demonstrate the effectiveness and efficiency of our DRO approach.

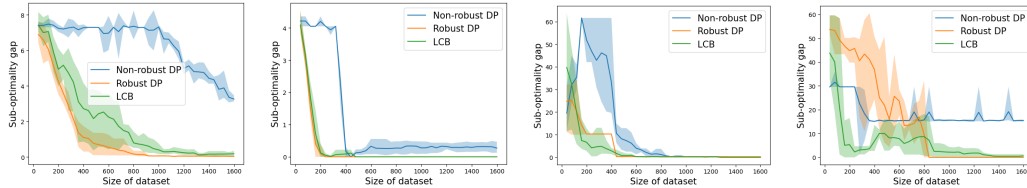

(a) Garnet Problem under global coverage (b) Garnet Problem under Partial coverage (c) Frozen-Lake under Partial coverage (d) Frozen-Lake under Partial coverage

Figure 1: Sub-optimality gaps of Robust DP, LCB approach, and Non-robust DP.

## 5 CONCLUSION

In this paper, we revisit the problem of offline reinforcement learning from a novel angle of distributional robustness. We develop a DRO-based approach to solve offline reinforcement learning. Our approach directly incorporates conservatism in estimating the transition dynamics instead of penalizing the reward of less-visited state-action pairs. Our algorithms are based on the robust dynamic programming approach, which is computationally efficient. We focus on the challenging partial coverage setting, and develop two uncertainty sets: the Hoeffding-style and the less conservative Bernstein-style. For the Hoeffding-style uncertainty set, we theoretically characterize its sample complexity, and show it matches with the best one of the LCB-based approaches using the Hoeffding-style bonus term. For the Bernstein-style uncertainty set, we show its sample complexity is minimax optimal. Our results provide a DRO-based framework to efficiently and effectively solve the problem of offline reinforcement learning.

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
