# A   NOTATIONS

We first introduce some notations that are used in our proofs. We denote the numbers of states and actions by $S, A$, i.e., $|\mathcal{S}| = S$, $|\mathcal{A}| = A$. For a transition kernel $\mathsf{P}$ and a policy $\pi$, $\mathsf{P}^\pi$ denotes the transition matrix induced by them, i.e., $\mathsf{P}^\pi(s) = \sum_a \pi(a|s)\mathsf{P}_s^a \in \Delta(\mathcal{S})$.

For any vector $V \in \mathbb{R}^S$, $V \circ V \in \mathbb{R}^S$ denotes the entry-wise multiplication, i.e., $V \circ V(s) = V(s) * V(s)$. For a distribution $q \in \Delta(\mathcal{S})$, it is straightforward to verify that the variance of $V$ w.r.t. $q$ can be rewritten as $\mathbf{Var}_q(V) = q(V \circ V) - (qV)^2$.

# B   A STRAIGHTFORWARD ANALYSIS: HOEFFDING'S INEQUALITY

**Lemma 2.** *With probability at least $1 - \delta$, it holds that for any $s, a$, $\mathsf{P}_s^a \in \hat{\mathcal{P}}_s^a$.*

*Proof.* If $2 < \sqrt{\frac{\log \frac{SA}{\delta}}{2N(s,a)}}$ (including the case $N(s,a) = 0$), the statement holds naturally due to the fact $\|p - q\| \le 2$ for any $p, q \in \Delta(\mathcal{S})$.

If $2 \ge \sqrt{\frac{\log \frac{SA}{\delta}}{2N(s,a)}}$, first by Hoeffding's inequality (Liu et al., 2022), we have that

$$\mathbb{P}(\|\hat{\mathsf{P}}_s^a - \mathsf{P}_s^a\| \ge k) \le \exp\left(-2N(s,a)k^2\right). \tag{10}$$

Plugging in $k = \sqrt{\frac{\log \frac{SA}{\delta}}{2N(s,a)}}$, we have that

$$\mathbb{P}\left(\|\hat{\mathsf{P}}_s^a - \mathsf{P}_s^a\| \ge \sqrt{\frac{\log \frac{SA}{\delta}}{2N(s,a)}}\right) \le \exp\left(-2N(s,a)\frac{\log \frac{SA}{\delta}}{2N(s,a)}\right) = \frac{\delta}{SA}, \tag{11}$$

which indicates the probability of $\mathsf{P}_s^a \in \hat{\mathcal{P}}_s^a$ for any $(s,a)$-pair is larger than $1 - \frac{\delta}{SA}$. Hence the probability of $\mathsf{P}_s^a \in \hat{\mathcal{P}}_s^a$ holding simultaneously for any $(s,a)$-pair is at least $1 - \delta$, which completes the proof. $\qquad\square$

**Theorem 3.** *(Restatement of Thm 1) With probability at least $1 - 2\delta$, it holds that*

$$V_{\mathsf{P}}^{\pi^*}(\rho) - V_{\mathsf{P}}^{\pi_r}(\rho) \le \frac{2}{(1-\gamma)^2}\frac{8SC^{\pi^*}\log\frac{NS}{\delta}}{N} + \frac{2}{(1-\gamma)^2}\sqrt{\frac{24SC^{\pi^*}\log\frac{SA}{\delta}}{N}}, \tag{12}$$

*To obtain an $\epsilon$-optimal policy, a dataset of size*

$$N = \mathcal{O}\left(\frac{SC^{\pi^*}}{(1-\gamma)^4\epsilon^2}\right) \tag{13}$$

*is required.*

*Proof.* In the following proof, we only focus on the case when

$$N > \frac{8SC^{\pi^*}\log\frac{NS}{\delta}}{1-\gamma}; \tag{14}$$

Otherwise, eq. (6) follows directly from the trivial bound $V_{\mathsf{P}}^{\pi^*}(\rho) - V_{\mathsf{P}}^{\pi_r}(\rho) \le \frac{1}{1-\gamma}$.

According to Lemma 1, with probability at least $1 - \delta$, $\mathsf{P} \in \hat{\mathcal{P}}$. Moreover, due to the fact $r(s,a) \ge \hat{r}(s,a)$, hence

$$V_{\mathsf{P}}^\pi(s) \ge V_{\hat{r},\mathsf{P}}^\pi(s) \ge V_{\hat{r},\hat{\mathcal{P}}}^\pi(s) = V_{\hat{\mathcal{P}}}^\pi(s) \tag{15}$$

for any $\pi$ and $s \in \mathcal{S}$, where $V_{\hat{r},\mathsf{P}}^\pi$ denotes the value function w.r.t. $\mathsf{P}$ and reward $\hat{r}$. Thus $V_{\mathsf{P}}^\pi \ge V_{\hat{\mathcal{P}}}^\pi$ for any policy $\pi$.

Therefore,

$$
\begin{aligned}
& V_{\mathsf{P}}^{\pi^*}(s) - V_{\mathsf{P}}^{\pi_r}(s) \\
& = V_{\mathsf{P}}^{\pi^*}(s) - V_{\hat{\mathcal{P}}}^{\pi_r}(s) + V_{\hat{\mathcal{P}}}^{\pi_r}(s) - V_{\mathsf{P}}^{\pi_r}(s) \\
& \leq V_{\mathsf{P}}^{\pi^*}(s) - V_{\hat{\mathcal{P}}}^{\pi_r}(s) \\
& = r(s, \pi^*(s)) + \gamma \mathsf{P}_s^{\pi^*(s)} V_{\mathsf{P}}^{\pi^*} - V_{\hat{\mathcal{P}}}^{\pi_r}(s) \\
& \overset{(a)}{\leq} r(s, \pi^*(s)) - \hat{r}(s, \pi^*(s)) + \gamma \mathsf{P}_s^{\pi^*(s)} V_{\mathsf{P}}^{\pi^*} - \gamma \sigma_{\hat{\mathcal{P}}_s^{\pi^*(s)}}(V_{\hat{\mathcal{P}}}^{\pi_r}) \\
& = r(s, \pi^*(s)) - \hat{r}(s, \pi^*(s)) + \gamma \mathsf{P}_s^{\pi^*(s)} V_{\mathsf{P}}^{\pi^*} - \gamma \mathsf{P}_s^{\pi^*(s)} V_{\hat{\mathcal{P}}}^{\pi_r} + \gamma \mathsf{P}_s^{\pi^*(s)} V_{\hat{\mathcal{P}}}^{\pi_r} - \gamma \sigma_{\hat{\mathcal{P}}_s^{\pi^*(s)}}(V_{\hat{\mathcal{P}}}^{\pi_r}) \\
& = r(s, \pi^*(s)) - \hat{r}(s, \pi^*(s)) + \gamma \mathsf{P}_s^{\pi^*(s)}(V_{\mathsf{P}}^{\pi^*} - V_{\hat{\mathcal{P}}}^{\pi_r}) + \gamma(\mathsf{P}_s^{\pi^*(s)} V_{\hat{\mathcal{P}}}^{\pi_r} - \sigma_{\hat{\mathcal{P}}_s^{\pi^*(s)}}(V_{\hat{\mathcal{P}}}^{\pi_r})) \\
& \triangleq \gamma \mathsf{P}_s^{\pi^*(s)}(V_{\mathsf{P}}^{\pi^*} - V_{\hat{\mathcal{P}}}^{\pi_r}) + b^*(V_{\hat{\mathcal{P}}}^{\pi_r}), \tag{16}
\end{aligned}
$$

where $(a)$ is from $V_{\hat{\mathcal{P}}}^{\pi_r}(s) = \max_a Q_{\hat{\mathcal{P}}}^{\pi_r}(s,a) \geq Q_{\hat{\mathcal{P}}}^{\pi_r}(s, \pi^*(s)) = \hat{r}(s, \pi^*(s)) + \gamma \sigma_{\hat{\mathcal{P}}_s^{\pi^*(s)}}(V_{\hat{\mathcal{P}}}^{\pi_r})$, and $b^*(V)(s) \triangleq r(s, \pi^*(s)) - \hat{r}(s, \pi^*(s)) + \gamma \mathsf{P}_s^{\pi^*(s)} V - \gamma \sigma_{\hat{\mathcal{P}}_s^{\pi^*(s)}}(V)$.

Recursively applying this inequality further implies

$$
V_{\mathsf{P}}^{\pi^*}(\rho) - V_{\mathsf{P}}^{\pi_r}(\rho) \leq \frac{1}{1-\gamma} \left\langle d^{\pi^*}, b^*(V_{\hat{\mathcal{P}}}^{\pi_r}) \right\rangle, \tag{17}
$$

where $d^{\pi^*}(\cdot) = (1-\gamma) \sum_{t=0}^{\infty} \gamma^t \mathbb{P}(S_t = \cdot | S_0 \sim \rho, \pi^*, \mathsf{P})$ is the discounted visitation distribution induced by $\pi^*$ and $\mathsf{P}$.

To bound the term in eq. (16), we introduce the following notations.

$$
\mathcal{S}_s \triangleq \left\{ s : N\mu(s, \pi^*(s)) \leq 8 \log \frac{NS}{\delta} \right\}, \tag{18}
$$

$$
\mathcal{S}_l \triangleq \left\{ s : N\mu(s, \pi^*(s)) > 8 \log \frac{NS}{\delta} \right\}. \tag{19}
$$

For $s \in \mathcal{S}_s$, from eq. (14), we have that

$$
\min \left\{ d^{\pi^*}(s), \frac{1}{S} \right\} \leq C^{\pi^*} \mu(s, \pi^*(s)) \leq \frac{8 C^{\pi^*} \log \frac{NS}{\delta}}{N} < \frac{1}{S}, \tag{20}
$$

which further implies that $d^{\pi^*}(s) \leq \frac{8 C^{\pi^*} \log \frac{NS}{\delta}}{N}$. Hence

$$
\sum_{s \in \mathcal{S}_s} d^{\pi^*}(s) b^*(V_{\hat{\mathcal{P}}}^{\pi_r})(s) \leq \frac{2}{1-\gamma} \frac{8 S C^{\pi^*} \log \frac{NS}{\delta}}{N}, \tag{21}
$$

which is due to $b^*(V)(s) \triangleq r(s, \pi^*(s)) - \hat{r}(s, \pi^*(s)) + \gamma \mathsf{P}_s^{\pi^*(s)} V - \gamma \sigma_{\hat{\mathcal{P}}_s^{\pi^*(s)}}(V) \leq 1 + \frac{\gamma}{1-\gamma} \leq \frac{2}{1-\gamma}$.

We then consider $s \in \mathcal{S}_l$. From the definition, $N\mu(s, \pi^*(s)) > 8 \log \frac{NS}{\delta}$. According to Lemma 5, with probability $1-\delta$,

$$
\max\{12 N(s, \pi^*(s)), 8 \log \frac{NS}{\delta}\} \geq N\mu(s, \pi^*(s)) > 8 \log \frac{NS}{\delta}, \tag{22}
$$

hence $\max\{12 N(s, \pi^*(s)), 8 \log \frac{NS}{\delta}\} = 12 N(s, \pi^*(s))$ and $N(s, \pi^*(s)) \geq \frac{2}{3} \log \frac{NS}{\delta} > 0$. This hence implies that for any $s \in \mathcal{S}_l$, $N(s, \pi^*(s)) > 0$ and $\hat{r}(s, \pi^*(s)) = r(s, \pi^*(s))$. Thus

$$
\begin{aligned}
|b^*(V_{\hat{\mathcal{P}}}^{\pi_r})(s)| & = \gamma |\mathsf{P}_s^{\pi^*(s)} V_{\hat{\mathcal{P}}}^{\pi_r} - \sigma_{\hat{\mathcal{P}}_s^{\pi^*(s)}}(V_{\hat{\mathcal{P}}}^{\pi_r})| \\
& \leq \|\mathsf{P}_s^{\pi^*(s)} - \mathsf{Q}_s^{\pi^*(s)}\|_1 \|V_{\hat{\mathcal{P}}}^{\pi_r}\|_{\infty} \\
& \leq \frac{2}{1-\gamma} \min \left\{ 2, \sqrt{\frac{\log \frac{SA}{\delta}}{2 N(s, \pi^*(s))}} \right\}
\end{aligned}
$$

$$\leq \frac{1}{1-\gamma} \sqrt{\frac{2\log\frac{SA}{\delta}}{N(s,\pi^*(s))}}. \tag{23}$$

Moreover, from eq. (22),

$$\frac{1}{N(s,\pi^*(s))} \leq \frac{12}{N\mu(s,\pi^*(s))} \leq \frac{12C^{\pi^*}}{N\min\{d^{\pi^*}(s),\frac{1}{S}\}} \leq \frac{12C^{\pi^*}}{N}\left(\frac{1}{d^{\pi^*}(s)} + S\right). \tag{24}$$

Combining with eq. (23) further implies

$$|b^*(V_{\hat{\mathcal{P}}}^{\pi_r})(s)| \leq \frac{1}{1-\gamma}\sqrt{2\log\frac{SA}{\delta}}\sqrt{\frac{12C^{\pi^*}}{N}\left(\frac{1}{d^{\pi^*}(s)},S\right)}$$

$$\leq \frac{1}{1-\gamma}\sqrt{2\log\frac{SA}{\delta}}\sqrt{\frac{12C^{\pi^*}}{N}\frac{1}{d^{\pi^*}(s)}} + \frac{1}{1-\gamma}\sqrt{2\log\frac{SA}{\delta}}\sqrt{\frac{12C^{\pi^*}}{N}S}. \tag{25}$$

Thus

$$\sum_{s\in\mathcal{S}_l} d^{\pi^*}(s)b^*(V_{\hat{\mathcal{P}}}^{\pi_r})(s) \leq \frac{1}{1-\gamma}\sqrt{\frac{24C^{\pi^*}\log\frac{SA}{\delta}}{N}}\sum_s \sqrt{d^{\pi^*}(s)} + \frac{1}{1-\gamma}\sqrt{\frac{24SC^{\pi^*}\log\frac{SA}{\delta}}{N}}$$

$$\leq \frac{1}{1-\gamma}\sqrt{\frac{24SC^{\pi^*}\log\frac{2}{\delta}}{N}} + \frac{1}{1-\gamma}\sqrt{\frac{24SC^{\pi^*}\log\frac{SA}{\delta}}{N}}$$

$$= \frac{2}{1-\gamma}\sqrt{\frac{24SC^{\pi^*}\log\frac{SA}{\delta}}{N}}. \tag{26}$$

Thus combining eq. (21) and eq. (26) implies

$$V_{\mathsf{P}}^{\pi^*}(\rho) - V_{\mathsf{P}}^{\pi_r}(\rho)$$
$$\leq \frac{1}{1-\gamma}\left\langle d^{\pi^*}, b^*(V_{\hat{\mathcal{P}}}^{\pi_r})\right\rangle$$
$$\leq \frac{2}{(1-\gamma)^2}\frac{8SC^{\pi^*}\log\frac{NS}{\delta}}{N} + \frac{2}{(1-\gamma)^2}\sqrt{\frac{24SC^{\pi^*}\log\frac{SA}{\delta}}{N}}, \tag{27}$$

which completes the proof. $\square$

## C A REFINED ANALYSIS: BERNSTEIN'S INEQUALITY

In this section, we provide a refined analysis of the sub-optimality gap using Bernstein's Inequality.

**Theorem 4.** *(Restatement of Thm 2) Consider the robust MDP $\hat{\mathcal{M}}$, there exist an optimal policy $\pi_r$ such that there exists some universal constants $K_1, K_2$, such that with probability at least $1-4\delta$, if $N \geq \frac{1}{(1-\gamma)\mu_{\min}^2}$, it holds that*

$$V_{\mathsf{P}}^{\pi^*}(\rho) - V_{\mathsf{P}}^{\pi_r}(\rho) \leq \frac{SK_1C^{\pi^*}\log\frac{NS}{\delta}}{(1-\gamma)^2N} + \sqrt{\frac{K_2SC^{\pi^*}\log\frac{4N}{\delta}}{(1-\gamma)^3N}} + \frac{384\log^2\frac{4SAN}{(1-\gamma)\delta}}{(1-\gamma)^2N\mu_{\min}}, \tag{28}$$

*where $K_1 = 2c_2 + 80c_1$, $K_2 = 442368$. For any $\epsilon$, the sample complexity required to obtain an $\epsilon$-optimal policy is*

$$\mathcal{O}\left(\frac{SC^{\pi^*}}{(1-\gamma)^3\epsilon^2} + \frac{1}{(1-\gamma)\mu_{\min}^2}\right). \tag{29}$$

*Proof.* We first have that

$$V_{\mathsf{P}}^{\pi^*} - V_{\mathsf{P}}^{\pi_r} = \underbrace{V_{\mathsf{P}}^{\pi^*} - V_{\hat{\mathcal{P}}}^{\pi_r}}_{\Delta_1} + \underbrace{V_{\hat{\mathcal{P}}}^{\pi_r} - V_{\mathsf{P}}^{\pi_r}}_{\Delta_2}. \tag{30}$$

In the following proof, we only focus on the case when

$$N > \max \left\{ \frac{SK_1 C^{\pi^*} \log \frac{NS}{\delta}}{1 - \gamma}, \frac{8 \log \frac{4SA}{\delta}}{(1 - \gamma)\mu_{\min}^2} \right\};$$

(31)

Otherwise, eq. (8) follows directly from the trivial bound $V_{\mathsf{P}}^{\pi^*}(\rho) - V_{\mathsf{P}}^{\pi_r}(\rho) \leq \frac{1}{1-\gamma}$.

We note that Lemma 8 of (Shi & Chi, 2022) states that under eq. (31), with probability $1 - \delta$, for any $(s, a)$ pair,

$$N(s, a) \geq \frac{N\mu(s, a)}{8 \log \frac{4SA}{\delta}}.$$

(32)

This moreover implies that with probability $1 - \delta$, if $\mu(s, a) > 0$, then $N(s, a) > 0$. We hence focus on the case when this event holds.

The remaining proof can be completed by combining the following two lemmas. $\qquad \square$

**Theorem 5.** *With probability at least $1 - 2\delta$, it holds that*

$$\rho^\top \Delta_1 \leq \frac{2c_2 S C^{\pi^*} \log \frac{4N}{\delta}}{(1 - \gamma)^2 N} + \frac{80S c_1 C^{\pi^*} \log \frac{NS}{\delta}}{(1 - \gamma)^2 N} + \frac{96}{\gamma} \sqrt{\frac{48S C^{\pi^*} \log \frac{4N}{\delta}}{(1 - \gamma)^3 N}}.$$

(33)

*Proof.* We first define the following set:

$$\mathcal{S}_0 \triangleq \{s : d^{\pi^*}(s) = 0\}.$$

(34)

And for $s \notin \mathcal{S}_0$, it holds that $d^{\pi^*}(s) > 0$.

We first consider $s \notin \mathcal{S}_0$. Due to the fact $d^{\pi^*}(s) > 0$, hence $d^{\pi^*}(s, \pi^*(s)) = d^{\pi^*}(s)\pi^*(\pi^*(s)|s) > 0$. Thus it implies that $\mu(s, \pi^*(s)) > 0$, and equation 32 further implies $N(s, \pi^*(s)) > 0$ and $\hat{r}(s, \pi^*(s)) = r(s, \pi^*(s))$. Hence we have that

$$V_{\mathsf{P}}^{\pi^*}(s) - V_{\hat{\mathcal{P}}}^{\pi_r}(s) = r(s, \pi^*(s)) + \gamma \mathsf{P}_s^{\pi^*(s)} V_{\mathsf{P}}^{\pi^*} - V_{\hat{\mathcal{P}}}^{\pi_r}(s)$$

$$\overset{(a)}{\leq} \gamma \mathsf{P}_s^{\pi^*(s)} V_{\mathsf{P}}^{\pi^*} - \gamma \sigma_{\hat{\mathcal{P}}_s^{\pi^*(s)}}(V_{\hat{\mathcal{P}}}^{\pi_r})$$

$$= \gamma \mathsf{P}_s^{\pi^*(s)} V_{\mathsf{P}}^{\pi^*} - \gamma \mathsf{P}_s^{\pi^*(s)} V_{\hat{\mathcal{P}}}^{\pi_r} + \gamma \mathsf{P}_s^{\pi^*(s)} V_{\hat{\mathcal{P}}}^{\pi_r} - \gamma \sigma_{\hat{\mathcal{P}}_s^{\pi^*(s)}}(V_{\hat{\mathcal{P}}}^{\pi_r})$$

$$= \gamma \mathsf{P}_s^{\pi^*(s)} (V_{\mathsf{P}}^{\pi^*} - V_{\hat{\mathcal{P}}}^{\pi_r}) + \gamma (\mathsf{P}_s^{\pi^*(s)} V_{\hat{\mathcal{P}}}^{\pi_r} - \sigma_{\hat{\mathcal{P}}_s^{\pi^*(s)}}(V_{\hat{\mathcal{P}}}^{\pi_r})), \quad (35)$$

where $(a)$ is from $V_{\hat{\mathcal{P}}}^{\pi_r}(s) = \max_a Q_{\hat{\mathcal{P}}}^{\pi_r}(s, a) \geq Q_{\hat{\mathcal{P}}}^{\pi_r}(s, \pi^*(s)) = \hat{r}(s, \pi^*(s)) + \gamma \sigma_{\hat{\mathcal{P}}_s^{\pi^*(s)}}(V_{\hat{\mathcal{P}}}^{\pi_r}) = r(s, \pi^*(s)) + \gamma \sigma_{\hat{\mathcal{P}}_s^{\pi^*(s)}}(V_{\hat{\mathcal{P}}}^{\pi_r})$.

For $s \in \mathcal{S}_0$, it holds that

$$V_{\mathsf{P}}^{\pi^*}(s) - V_{\hat{\mathcal{P}}}^{\pi_r}(s) = r(s, \pi^*(s)) + \gamma \mathsf{P}_s^{\pi^*(s)} V_{\mathsf{P}}^{\pi^*} - V_{\hat{\mathcal{P}}}^{\pi_r}(s)$$

$$\overset{(a)}{\leq} \gamma \mathsf{P}_s^{\pi^*(s)} V_{\mathsf{P}}^{\pi^*} - \gamma \sigma_{\hat{\mathcal{P}}_s^{\pi^*(s)}}(V_{\hat{\mathcal{P}}}^{\pi_r}) + r(s, \pi^*(s)) - \hat{r}(s, \pi^*(s))$$

$$\leq \gamma \mathsf{P}_s^{\pi^*(s)} (V_{\mathsf{P}}^{\pi^*} - V_{\hat{\mathcal{P}}}^{\pi_r}) + \gamma (\mathsf{P}_s^{\pi^*(s)} V_{\hat{\mathcal{P}}}^{\pi_r} - \sigma_{\hat{\mathcal{P}}_s^{\pi^*(s)}}(V_{\hat{\mathcal{P}}}^{\pi_r})) + 1, \quad (36)$$

where $(a)$ follows similarly from eq. (35), and the last inequality is from $r(s, \pi^*(s)) \leq 1$.

Hence combining eq. (35) and eq. (36) implies

$$V_{\mathsf{P}}^{\pi^*}(s) - V_{\hat{\mathcal{P}}}^{\pi_r}(s) \leq \gamma \mathsf{P}_s^{\pi^*(s)} (V_{\mathsf{P}}^{\pi^*} - V_{\hat{\mathcal{P}}}^{\pi_r}) + b^*(V)(s),$$

(37)

where $b^*(V)(s) \triangleq \gamma \mathsf{P}_s^{\pi^*(s)} V - \gamma \sigma_{\hat{\mathcal{P}}_s^{\pi^*(s)}}(V)$ if $s \notin \mathcal{S}_0$, and $b^*(V)(s) \triangleq 1 + \gamma \mathsf{P}_s^{\pi^*(s)} V - \gamma \sigma_{\hat{\mathcal{P}}_s^{\pi^*(s)}}(V)$ for $s \in \mathcal{S}_0$.

Moreover we set

$$\tilde{b}(V_{\hat{\mathcal{P}}}^{\pi_r})(s) = \max\{0, b^*(V_{\hat{\mathcal{P}}}^{\pi_r})(s)\},$$

(38)

then it holds that $b^*(V_{\hat{\mathcal{P}}}^{\pi_r}) \leq \tilde{b}(V_{\hat{\mathcal{P}}}^{\pi_r})$.

Then apply eq. (37) recursively and we have that

$$\rho^\top \Delta_1 \leq \frac{1}{1-\gamma} \left\langle d^{\pi^*}, \tilde{b}(V_{\hat{\mathcal{P}}}^{\pi_r}) \right\rangle, \tag{39}$$

Not that for $s \in \mathcal{S}_0$, it holds that $d^{\pi^*}(s) = 0$, and

$$\left\langle d^{\pi^*}, \tilde{b}(V_{\hat{\mathcal{P}}}^{\pi_r}) \right\rangle = \sum_{s \notin \mathcal{S}_0} d^{\pi^*}(s)\tilde{b}(V_{\hat{\mathcal{P}}}^{\pi_r})(s). \tag{40}$$

This implies that we only need to focus on $s \notin \mathcal{S}_0$. We further defined the following sets:

$$\mathcal{S}_s \triangleq \left\{ s \notin \mathcal{S}_0 : N\mu(s, \pi^*(s)) \leq 8 \log \frac{NS}{\delta} \right\}, \tag{41}$$

$$\mathcal{S}_l \triangleq \left\{ s \notin \mathcal{S}_0 : N\mu(s, \pi^*(s)) > 8 \log \frac{NS}{\delta} \right\}. \tag{42}$$

For $s \in \mathcal{S}_s$, we have that

$$\min\left\{ d^{\pi^*}(s), \frac{1}{S} \right\} \leq C^{\pi^*}\mu(s, \pi^*(s)) \leq \frac{8C^{\pi^*}\log\frac{NS}{\delta}}{N} \overset{(a)}{<} \frac{1}{S}, \tag{43}$$

where $(a)$ is due to the fact eq. (31).

This further implies that $d^{\pi^*}(s) \leq \frac{8C^{\pi^*}\log\frac{NS}{\delta}}{N}$. Hence

$$\sum_{s \in \mathcal{S}_s} d^{\pi^*}(s)\tilde{b}(V_{\hat{\mathcal{P}}}^{\pi_r})(s) \leq \frac{2}{1-\gamma}\frac{8SC^{\pi^*}\log\frac{NS}{\delta}}{N}, \tag{44}$$

which is due to $\|\mathsf{P}_s^{\pi^*(s)}V_{\hat{\mathcal{P}}}^{\pi_r} - \sigma_{\hat{\mathcal{P}}_s^{\pi^*(s)}}(V_{\hat{\mathcal{P}}}^{\pi_r})\| \leq \frac{2}{1-\gamma}$.

We then consider $s \in \mathcal{S}_l$. Note that from the definition and equation 32, it holds that $N(s, \pi^*(s)) > 0$ for $s \in \mathcal{S}_l$.

Therefore it holds that

$$\begin{aligned}
\tilde{b}(V_{\hat{\mathcal{P}}}^{\pi_r})(s) &\leq |\gamma\mathsf{P}_s^{\pi^*(s)}V_{\hat{\mathcal{P}}}^{\pi_r} - \gamma\sigma_{\hat{\mathcal{P}}_s^{\pi^*(s)}}(V_{\hat{\mathcal{P}}}^{\pi_r})| \\
&\leq |\gamma\mathsf{P}_s^{\pi^*(s)}V_{\hat{\mathcal{P}}}^{\pi_r} - \gamma\hat{\mathsf{P}}_s^{\pi^*(s)}V_{\hat{\mathcal{P}}}^{\pi_r}| + |\gamma\hat{\mathsf{P}}_s^{\pi^*(s)}V_{\hat{\mathcal{P}}}^{\pi_r} - \gamma\sigma_{\hat{\mathcal{P}}_s^{\pi^*(s)}}(V_{\hat{\mathcal{P}}}^{\pi_r})| \\
&\leq \gamma|\mathsf{P}_s^{\pi^*(s)}V_{\hat{\mathcal{P}}}^{\pi_r} - \hat{\mathsf{P}}_s^{\pi^*(s)}V_{\hat{\mathcal{P}}}^{\pi_r}| + \sqrt{\frac{\log\frac{4N}{\delta}\mathbf{Var}_{\hat{\mathsf{P}}_s^{\pi^*(s)}}(V_{\hat{\mathcal{P}}}^{\pi_r})}{N(s, \pi^*(s))}},
\end{aligned} \tag{45}$$

where the last inequality is shown as follows.

Since $\hat{\mathsf{P}}_s^{\pi^*(s)} \in \hat{\mathcal{P}}_s^a$, we have that $\hat{\mathsf{P}}_s^{\pi^*(s)}V_{\hat{\mathcal{P}}}^a \geq \sigma_{\hat{\mathcal{P}}_s^{\pi^*(s)}}(V_{\hat{\mathcal{P}}}^{\pi_r})$. Now note that it is shown in Iyengar (2005); Shi et al. (2023) that

$$\sigma_{\hat{\mathcal{P}}_s^{\pi^*(s)}}(V_{\hat{\mathcal{P}}}^{\pi_r}) = \max_{\mu \in [0, V_{\hat{\mathcal{P}}}^{\pi_r}]} \left\{ \hat{\mathsf{P}}_s^{\pi^*(s)}(V_{\hat{\mathcal{P}}}^{\pi_r} - \mu) - \sqrt{R_s^{\pi^*(s)}\mathbf{Var}_{\hat{\mathsf{P}}_s^{\pi^*(s)}}(V_{\hat{\mathcal{P}}}^{\pi_r} - \mu)} \right\}. \tag{46}$$

Thus

$$\begin{aligned}
&|\gamma\hat{\mathsf{P}}_s^{\pi^*(s)}V_{\hat{\mathcal{P}}}^{\pi_r} - \gamma\sigma_{\hat{\mathcal{P}}_s^{\pi^*(s)}}(V_{\hat{\mathcal{P}}}^{\pi_r})| \\
&= \gamma\hat{\mathsf{P}}_s^{\pi^*(s)}V_{\hat{\mathcal{P}}}^{\pi_r} - \gamma\sigma_{\hat{\mathcal{P}}_s^{\pi^*(s)}}(V_{\hat{\mathcal{P}}}^{\pi_r}) \\
&= \gamma\hat{\mathsf{P}}_s^{\pi^*(s)}V_{\hat{\mathcal{P}}}^{\pi_r} - \gamma\max_{\mu \in [0, V_{\hat{\mathcal{P}}}^{\pi_r}]}\left\{ \hat{\mathsf{P}}_s^{\pi^*(s)}(V_{\hat{\mathcal{P}}}^{\pi_r} - \mu) - \sqrt{R_s^{\pi^*(s)}\mathbf{Var}_{\hat{\mathsf{P}}_s^{\pi^*(s)}}(V_{\hat{\mathcal{P}}}^{\pi_r} - \mu)} \right\} \\
&\overset{(a)}{\leq} \gamma\hat{\mathsf{P}}_s^{\pi^*(s)}V_{\hat{\mathcal{P}}}^{\pi_r} - \gamma\left( \hat{\mathsf{P}}_s^{\pi^*(s)}(V_{\hat{\mathcal{P}}}^{\pi_r}) - \sqrt{R_s^{\pi^*(s)}\mathbf{Var}_{\hat{\mathsf{P}}_s^{\pi^*(s)}}(V_{\hat{\mathcal{P}}}^{\pi_r})} \right)
\end{aligned}$$

$$= \sqrt{R_s^{\pi^*(s)} \mathbf{Var}_{\hat{\mathsf{P}}_s^{\pi^*(s)}}(V_{\hat{\mathcal{P}}}^{\pi_r})}, \tag{47}$$

where $(a)$ is due to the maximum term is larger than the function value at $\mu = 0$, and this inequality completes the proof of Equation (45).

To further bound eq. (45), we invoke Lemma 7 and have that

$$\left| \mathsf{P}_s^{\pi^*(s)} V_{\hat{\mathcal{P}}}^{\pi_r} - \hat{\mathsf{P}}_s^{\pi^*(s)} V_{\hat{\mathcal{P}}}^{\pi_r} \right|$$

$$\leq 12 \sqrt{\frac{\mathbf{Var}_{\hat{\mathsf{P}}_s^{\pi^*(s)}}(V_{\hat{\mathcal{P}}}^{\pi_r}) \log \frac{4N}{\delta}}{N(s, \pi^*(s))}} + \frac{74 \log \frac{4N}{\delta}}{(1-\gamma)N(s, \pi^*(s))}$$

$$\leq 12 \sqrt{\frac{\log \frac{4N}{\delta}}{N(s, \pi^*(s))} \left( 2\mathbf{Var}_{\mathsf{P}_s^{\pi^*(s)}}(V_{\hat{\mathcal{P}}}^{\pi_r}) + \frac{41 \log \frac{4N}{\delta}}{(1-\gamma)^2 N(s, \pi^*(s))} \right)} + \frac{74 \log \frac{4N}{\delta}}{(1-\gamma)N(s, \pi^*(s))}$$

$$\leq 12 \sqrt{\frac{2 \log \frac{4N}{\delta}}{N(s, \pi^*(s))} \mathbf{Var}_{\mathsf{P}_s^{\pi^*(s)}}(V_{\hat{\mathcal{P}}}^{\pi_r})} + \frac{(74 + 12\sqrt{41}) \log \frac{4N}{\delta}}{(1-\gamma)N(s, \pi^*(s))}, \tag{48}$$

where the last inequality is from $\sqrt{x+y} \leq \sqrt{x} + \sqrt{y}$.

Combine eq. (45) and eq. (48), we further have that

$$\tilde{b}(V_{\hat{\mathcal{P}}}^{\pi_r})(s) \leq 24 \sqrt{\frac{2 \log \frac{4N}{\delta}}{N(s, \pi^*(s))} \mathbf{Var}_{\mathsf{P}_s^{\pi^*(s)}}(V_{\hat{\mathcal{P}}}^{\pi_r})} + \frac{(74 + 12\sqrt{41}) \log \frac{4N}{\delta}}{(1-\gamma)N(s, \pi^*(s))} + \frac{\log \frac{N}{\delta}}{(1-\gamma)N(s, \pi^*(s))}$$

$$\leq 24 \sqrt{\frac{2 \log \frac{4N}{\delta}}{N(s, \pi^*(s))} \mathbf{Var}_{\mathsf{P}_s^{\pi^*(s)}}(V_{\hat{\mathcal{P}}}^{\pi_r})} + \frac{c_1 \log \frac{4N}{\delta}}{(1-\gamma)N(s, \pi^*(s))}, \tag{49}$$

where $c_1 = 75 + 12\sqrt{41}$.

Note that in eq. (24), we showed that $\frac{1}{N(s, \pi^*(s))} \leq \frac{12C^{\pi^*}}{N}(S + \frac{1}{d^{\pi^*}(s)})$. Hence plugging in eq. (49) implies that

$$\tilde{b}(V_{\hat{\mathcal{P}}}^{\pi_r})(s) \leq 24 \sqrt{\frac{24C^{\pi^*} \log \frac{4N}{\delta}}{N} \mathbf{Var}_{\mathsf{P}_s^{\pi^*(s)}}(V_{\hat{\mathcal{P}}}^{\pi_r})} \left( \sqrt{S} + \frac{1}{\sqrt{d^{\pi^*}(s)}} \right)$$

$$+ \frac{12c_1 C^{\pi^*} \log \frac{4N}{\delta}}{(1-\gamma)N} \left( S + \frac{1}{d^{\pi^*}(s)} \right). \tag{50}$$

Firstly we have that

$$\sum_{s \in \mathcal{S}_l} 24 d^{\pi^*}(s) \sqrt{\frac{24C^{\pi^*} \log \frac{4N}{\delta}}{N} \mathbf{Var}_{\mathsf{P}_s^{\pi^*(s)}}(V_{\hat{\mathcal{P}}}^{\pi_r})} \left( \sqrt{S} + \frac{1}{\sqrt{d^{\pi^*}(s)}} \right)$$

$$= \sum_{s \in \mathcal{S}_l} 24 d^{\pi^*}(s) \sqrt{\frac{24SC^{\pi^*} \log \frac{4N}{\delta}}{N} \mathbf{Var}_{\mathsf{P}_s^{\pi^*(s)}}(V_{\hat{\mathcal{P}}}^{\pi_r})} + \sum_{s \in \mathcal{S}_l} 12 \sqrt{d^{\pi^*}(s)} \sqrt{\frac{24C^{\pi^*} \log \frac{4N}{\delta}}{N} \mathbf{Var}_{\mathsf{P}_s^{\pi^*(s)}}(V_{\hat{\mathcal{P}}}^{\pi_r})}$$

$$= 24 \sqrt{\frac{24C^{\pi^*} \log \frac{4N}{\delta}}{N}} \left( \sum_{s \in \mathcal{S}_l} \sqrt{d^{\pi^*}(s) \mathbf{Var}_{\mathsf{P}_s^{\pi^*(s)}}(V_{\hat{\mathcal{P}}}^{\pi_r})} + \sum_{s \in \mathcal{S}_l} \sqrt{d^{\pi^*}(s)} \sqrt{S d^{\pi^*}(s) \mathbf{Var}_{\mathsf{P}_s^{\pi^*(s)}}(V_{\hat{\mathcal{P}}}^{\pi_r})} \right)$$

$$\overset{(a)}{\leq} 24 \sqrt{\frac{24C^{\pi^*} \log \frac{4N}{\delta}}{N}} \left( \sqrt{S} \sqrt{\sum_{s \in \mathcal{S}_l} d^{\pi^*}(s) \mathbf{Var}_{\mathsf{P}_s^{\pi^*(s)}}(V_{\hat{\mathcal{P}}}^{\pi_r})} + \sqrt{\sum_{s \in \mathcal{S}_l} S d^{\pi^*}(s) \mathbf{Var}_{\mathsf{P}_s^{\pi^*(s)}}(V_{\hat{\mathcal{P}}}^{\pi_r})} \right)$$

$$= 48 \sqrt{\frac{24SC^{\pi^*} \log \frac{4N}{\delta}}{N}} \sqrt{\sum_{s \in \mathcal{S}_l} d^{\pi^*}(s) \mathbf{Var}_{\mathsf{P}_s^{\pi^*(s)}}(V_{\hat{\mathcal{P}}}^{\pi_r})}, \tag{51}$$

where $(a)$ is from Cauchy's inequality and the fact $\sum_{s \in \mathcal{S}_l} d^{\pi^*}(s) \leq 1$.

In addition, we have that

$$\sum_{s \in \mathcal{S}_l} d^{\pi^*}(s) \frac{12 c_1 C^{\pi^*} \log \frac{4N}{\delta}}{(1-\gamma)N} \left( S + \frac{1}{d^{\pi^*}(s)} \right) \leq \frac{24 c_1 S C^{\pi^*} \log \frac{4N}{\delta}}{(1-\gamma)N}. \tag{52}$$

Combine the two inequalities above and we have that

$$\sum_{s \in \mathcal{S}_l} d^{\pi^*}(s) \tilde{b}(V_{\hat{\mathcal{P}}}^{\pi_r})(s)$$

$$\leq 48 \sqrt{\frac{24 S C^{\pi^*} \log \frac{4N}{\delta}}{N}} \sqrt{\sum_{s \in \mathcal{S}_l} d^{\pi^*}(s) \mathbf{Var}_{\mathsf{P}_s^{\pi^*(s)}}(V_{\hat{\mathcal{P}}}^{\pi_r})} + \frac{24 S c_1 C^{\pi^*} \log \frac{4N}{\delta}}{(1-\gamma)N}. \tag{53}$$

Then we combine eq. (44) and eq. (53), and it implies that

$$\langle d^{\pi^*}, \tilde{b}(V_{\hat{\mathcal{P}}}^{\pi_r}) \rangle$$

$$= \sum_{s \in \mathcal{S}_s} d^{\pi^*}(s) \tilde{b}(V_{\hat{\mathcal{P}}}^{\pi_r})(s) + \sum_{s \in \mathcal{S}_l} d^{\pi^*}(s) \tilde{b}(V_{\hat{\mathcal{P}}}^{\pi_r})(s)$$

$$\leq \frac{16 S C^{\pi^*} \log \frac{NS}{\delta}}{(1-\gamma)N} + 48 \sqrt{\frac{24 S C^{\pi^*} \log \frac{4N}{\delta}}{N}} \sqrt{\sum_{s \in \mathcal{S}_l} d^{\pi^*}(s) \mathbf{Var}_{\mathsf{P}_s^{\pi^*(s)}}(V_{\hat{\mathcal{P}}}^{\pi_r})} + \frac{24 S c_1 C^{\pi^*} \log \frac{4N}{\delta}}{(1-\gamma)N}$$

$$\leq \frac{40 c_1 S C^{\pi^*} \log \frac{NS}{\delta}}{(1-\gamma)N} + 48 \sqrt{\frac{24 S C^{\pi^*} \log \frac{4N}{\delta}}{N}} \sqrt{\sum_{s \in \mathcal{S}} d^{\pi^*}(s) \mathbf{Var}_{\mathsf{P}_s^{\pi^*(s)}}(V_{\hat{\mathcal{P}}}^{\pi_r})}. \tag{54}$$

We then bound the term $\sum_{s \in \mathcal{S}} d^{\pi^*}(s) \mathbf{Var}_{\mathsf{P}_s^{\pi^*(s)}}(V_{\hat{\mathcal{P}}}^{\pi_r})$. We first claim the following inequality:

$$V_{\hat{\mathcal{P}}}^{\pi_r} - \gamma \mathsf{P}^{\pi^*} V_{\hat{\mathcal{P}}}^{\pi_r} + 2\tilde{b}(V_{\hat{\mathcal{P}}}^{\pi_r}) \geq 0. \tag{55}$$

To prove eq. (55), we note that

$$\begin{aligned} V_{\hat{\mathcal{P}}}^{\pi_r}(s) &= \max_a Q_{\hat{\mathcal{P}}}^{\pi_r}(s, a) \\ &\geq Q_{\hat{\mathcal{P}}}^{\pi_r}(s, \pi^*(s)) \\ &= \hat{r}(s, \pi^*(s)) + \gamma \sigma_{\hat{\mathcal{P}}_s^{\pi^*(s)}}(V_{\hat{\mathcal{P}}}^{\pi_r}) \\ &= \hat{r}(s, \pi^*(s)) + \gamma \mathsf{P}_s^{\pi^*(s)} V_{\hat{\mathcal{P}}}^{\pi_r} - \gamma \mathsf{P}_s^{\pi^*(s)} V_{\hat{\mathcal{P}}}^{\pi_r} + \gamma \sigma_{\hat{\mathcal{P}}_s^{\pi^*(s)}}(V_{\hat{\mathcal{P}}}^{\pi_r}) \\ &\overset{(a)}{\geq} \hat{r}(s, \pi^*(s)) + \gamma \mathsf{P}_s^{\pi^*(s)} V_{\hat{\mathcal{P}}}^{\pi_r} - b^*(V_{\hat{\mathcal{P}}}^{\pi_r})(s) \\ &\geq \hat{r}(s, \pi^*(s)) + \gamma \mathsf{P}_s^{\pi^*(s)} V_{\hat{\mathcal{P}}}^{\pi_r} - 2\tilde{b}(V_{\hat{\mathcal{P}}}^{\pi_r})(s), \end{aligned} \tag{56}$$

where $(a)$ is from $b^*(V_{\hat{\mathcal{P}}}^{\pi_r})(s) \geq \gamma \mathsf{P}_s^{\pi^*(s)} V_{\hat{\mathcal{P}}}^{\pi_r} - \gamma \sigma_{\hat{\mathcal{P}}_s^{\pi^*(s)}}(V_{\hat{\mathcal{P}}}^{\pi_r})$.

Hence for any $s \in \mathcal{S}$,

$$V_{\hat{\mathcal{P}}}^{\pi_r}(s) - \gamma \mathsf{P}_s^{\pi^*(s)} V_{\hat{\mathcal{P}}}^{\pi_r} + 2\tilde{b}(V_{\hat{\mathcal{P}}}^{\pi_r})(s) \geq \hat{r}(s, \pi^*(s)) \geq 0, \tag{57}$$

which proves eq. (55).

Now with eq. (55), we first note that

$$\begin{aligned} (V_{\hat{\mathcal{P}}}^{\pi_r} \circ V_{\hat{\mathcal{P}}}^{\pi_r}) &- (\gamma \mathsf{P}^{\pi^*} V_{\hat{\mathcal{P}}}^{\pi_r}) \circ (\gamma \mathsf{P}^{\pi^*} V_{\hat{\mathcal{P}}}^{\pi_r}) \\ &= (V_{\hat{\mathcal{P}}}^{\pi_r} - \gamma \mathsf{P}^{\pi^*} V_{\hat{\mathcal{P}}}^{\pi_r}) \circ (V_{\hat{\mathcal{P}}}^{\pi_r} + \gamma \mathsf{P}^{\pi^*} V_{\hat{\mathcal{P}}}^{\pi_r}) \\ &\leq (V_{\hat{\mathcal{P}}}^{\pi_r} - \gamma \mathsf{P}^{\pi^*} V_{\hat{\mathcal{P}}}^{\pi_r} + 2\tilde{b}(V_{\hat{\mathcal{P}}}^{\pi_r})) \circ (V_{\hat{\mathcal{P}}}^{\pi_r} + \gamma \mathsf{P}^{\pi^*} V_{\hat{\mathcal{P}}}^{\pi_r}) \\ &\leq \frac{2}{1-\gamma}(V_{\hat{\mathcal{P}}}^{\pi_r} - \gamma \mathsf{P}^{\pi^*} V_{\hat{\mathcal{P}}}^{\pi_r} + 2\tilde{b}(V_{\hat{\mathcal{P}}}^{\pi_r})), \end{aligned} \tag{58}$$

where the last inequality is due to the fact $\|V_{\hat{\mathcal{P}}}^{\pi_r} + \gamma \mathsf{P}^{\pi^*} V_{\hat{\mathcal{P}}}^{\pi_r}\| \leq \frac{2}{1-\gamma}$ and eq. (55).

We then have that

$$
\begin{aligned}
&\sum_{s \in \mathcal{S}} d^{\pi^*}(s) \mathbf{Var}_{\mathsf{P}_s^{\pi^*(s)}}(V_{\hat{\mathcal{P}}}^{\pi_r}) \\
&= \langle d^{\pi^*}, \mathsf{P}^{\pi^*}(V_{\hat{\mathcal{P}}}^{\pi_r} \circ V_{\hat{\mathcal{P}}}^{\pi_r}) - (\mathsf{P}^{\pi^*} V_{\hat{\mathcal{P}}}^{\pi_r}) \circ (\mathsf{P}^{\pi^*} V_{\hat{\mathcal{P}}}^{\pi_r}) \rangle \\
&\overset{(a)}{\leq} \left\langle d^{\pi^*}, \mathsf{P}^{\pi^*}(V_{\hat{\mathcal{P}}}^{\pi_r} \circ V_{\hat{\mathcal{P}}}^{\pi_r}) - \frac{1}{\gamma^2}(V_{\hat{\mathcal{P}}}^{\pi_r} \circ V_{\hat{\mathcal{P}}}^{\pi_r}) + \frac{2}{\gamma^2(1-\gamma)}(V_{\hat{\mathcal{P}}}^{\pi_r} - \gamma \mathsf{P}^{\pi^*} V_{\hat{\mathcal{P}}}^{\pi_r} + 2\tilde{b}(V_{\hat{\mathcal{P}}}^{\pi_r})) \right\rangle \\
&\overset{(b)}{\leq} \left\langle d^{\pi^*}, \mathsf{P}^{\pi^*}(V_{\hat{\mathcal{P}}}^{\pi_r} \circ V_{\hat{\mathcal{P}}}^{\pi_r}) - \frac{1}{\gamma}(V_{\hat{\mathcal{P}}}^{\pi_r} \circ V_{\hat{\mathcal{P}}}^{\pi_r}) + \frac{2}{\gamma^2(1-\gamma)}(I - \gamma \mathsf{P}^{\pi^*})V_{\hat{\mathcal{P}}}^{\pi_r} + \frac{4}{\gamma^2(1-\gamma)}\tilde{b}(V_{\hat{\mathcal{P}}}^{\pi_r})) \right\rangle \\
&= \left\langle d^{\pi^*}, \frac{1}{\gamma}(\gamma \mathsf{P}^{\pi^*} - I)(V_{\hat{\mathcal{P}}}^{\pi_r} \circ V_{\hat{\mathcal{P}}}^{\pi_r}) + \frac{2}{\gamma^2(1-\gamma)}(I - \gamma \mathsf{P}^{\pi^*})V_{\hat{\mathcal{P}}}^{\pi_r} + \frac{4}{\gamma^2(1-\gamma)}\tilde{b}(V_{\hat{\mathcal{P}}}^{\pi_r})) \right\rangle \\
&= (d^{\pi^*})^\top (I - \gamma \mathsf{P}^{\pi^*})\left(-\frac{1}{\gamma}(V_{\hat{\mathcal{P}}}^{\pi_r} \circ V_{\hat{\mathcal{P}}}^{\pi_r}) + \frac{2}{\gamma^2(1-\gamma)}V_{\hat{\mathcal{P}}}^{\pi_r}\right) + \frac{4}{\gamma^2(1-\gamma)}\langle d^{\pi^*}, \tilde{b}(V_{\hat{\mathcal{P}}}^{\pi_r})\rangle \\
&\overset{(c)}{=} (1-\gamma)\rho^\top \left(-\frac{1}{\gamma}(V_{\hat{\mathcal{P}}}^{\pi_r} \circ V_{\hat{\mathcal{P}}}^{\pi_r}) + \frac{2}{\gamma^2(1-\gamma)}V_{\hat{\mathcal{P}}}^{\pi_r}\right) + \frac{4}{\gamma^2(1-\gamma)}\langle d^{\pi^*}, \tilde{b}(V_{\hat{\mathcal{P}}}^{\pi_r})\rangle \\
&\leq \frac{2}{\gamma^2}\rho^\top V_{\hat{\mathcal{P}}}^{\pi_r} + \frac{4}{\gamma^2(1-\gamma)}\langle d^{\pi^*}, \tilde{b}(V_{\hat{\mathcal{P}}}^{\pi_r})\rangle \\
&\leq \frac{2}{\gamma^2(1-\gamma)} + \frac{4}{\gamma^2(1-\gamma)}\langle d^{\pi^*}, \tilde{b}(V_{\hat{\mathcal{P}}}^{\pi_r})\rangle,
\end{aligned}
\tag{59}
$$

where $(a)$ is from eq. (58), $(b)$ is due to $\gamma < 1$, $(c)$ is from the definition of visitation distribution.

Hence by plugging eq. (59) in eq. (54), we have that

$$
\begin{aligned}
&\langle d^{\pi^*}, \tilde{b}(V_{\hat{\mathcal{P}}}^{\pi_r})\rangle \\
&\leq \frac{40 c_1 S C^{\pi^*} \log \frac{NS}{\delta}}{(1-\gamma)N} + 48\sqrt{\frac{24 S C^{\pi^*} \log \frac{4N}{\delta}}{N}}\sqrt{\sum_{s \in \mathcal{S}} d^{\pi^*}(s)\mathbf{Var}_{\mathsf{P}_s^{\pi^*(s)}}(V_{\hat{\mathcal{P}}}^{\pi_r})} \\
&\leq \frac{40 c_1 S C^{\pi^*} \log \frac{NS}{\delta}}{(1-\gamma)N} + 48\sqrt{\frac{24 S C^{\pi^*} \log \frac{4N}{\delta}}{N}}\sqrt{\frac{2}{\gamma^2(1-\gamma)} + \frac{4}{\gamma^2(1-\gamma)}\langle d^{\pi^*}, \tilde{b}(V_{\hat{\mathcal{P}}}^{\pi_r})\rangle} \\
&\leq \frac{40 c_1 S C^{\pi^*} \log \frac{NS}{\delta}}{(1-\gamma)N} + \frac{24}{\gamma}\sqrt{\frac{48 S C^{\pi^*} \log \frac{4N}{\delta}}{(1-\gamma)N}} + \frac{48}{\gamma}\sqrt{\frac{96 S C^{\pi^*} \log \frac{4N}{\delta}}{(1-\gamma)N}}\sqrt{\langle d^{\pi^*}, \tilde{b}(V_{\hat{\mathcal{P}}}^{\pi_r})\rangle} \\
&\overset{(a)}{\leq} \frac{1}{2}\langle d^{\pi^*}, \tilde{b}(V_{\hat{\mathcal{P}}}^{\pi_r})\rangle + c_2 \frac{S C^{\pi^*} \log \frac{4N}{\delta}}{(1-\gamma)N} + \frac{40 S c_1 C^{\pi^*} \log \frac{NS}{\delta}}{(1-\gamma)N} + \frac{48}{\gamma}\sqrt{\frac{48 S C^{\pi^*} \log \frac{4N}{\delta}}{(1-\gamma)N}},
\end{aligned}
\tag{60}
$$

where $(a)$ is from $x + y \geq 2\sqrt{xy}$ and $c_2 = 8 * 24^3 = 110592$. This inequality moreover implies that

$$
\langle d^{\pi^*}, \tilde{b}(V_{\hat{\mathcal{P}}}^{\pi_r})\rangle \leq 2c_2 \frac{S C^{\pi^*} \log \frac{4N}{\delta}}{(1-\gamma)N} + \frac{80 S c_1 C^{\pi^*} \log \frac{NS}{\delta}}{(1-\gamma)N} + \frac{96}{\gamma}\sqrt{\frac{48 S C^{\pi^*} \log \frac{4N}{\delta}}{(1-\gamma)N}}.
\tag{61}
$$

Recall the definition of $\Delta_1$, we hence have that

$$
\rho^\top \Delta_1 \leq \frac{2c_2 S C^{\pi^*} \log \frac{4N}{\delta}}{(1-\gamma)^2 N} + \frac{80 S c_1 C^{\pi^*} \log \frac{NS}{\delta}}{(1-\gamma)^2 N} + \frac{96}{\gamma}\sqrt{\frac{48 S C^{\pi^*} \log \frac{4N}{\delta}}{(1-\gamma)^3 N}}.
\tag{62}
$$

This hence completes the proof of the lemma. $\square$

**Theorem 6.** *With probability at least $1 - 2\delta$, it holds that*

$$
\rho^\top \Delta_2 \leq 2\sqrt{\frac{384 \log^2 \frac{4SAN}{(1-\gamma)\delta}}{(1-\gamma)^3 N^2 \mu_{\min}}} + 2\sqrt{\frac{384 \log^2 \frac{4SAN}{(1-\gamma)\delta}}{(1-\gamma)^3 N^3 \mu_{\min}}} + \frac{384 \log^2 \frac{4SAN}{(1-\gamma)\delta}}{(1-\gamma)^2 N \mu_{\min}}.
\tag{63}
$$

*Proof.* We first define the following set:

$$\mathcal{S}^0 \triangleq \{s \in \mathcal{S} : N(s) = 0\}. \tag{64}$$

Note that $N(s) = \sum_a N(s,a)$, hence it holds that $N(s,a) = 0$ for any $s \in \mathcal{S}^0$, $a \in \mathcal{A}$.

We moreover construct an absorbing MDP $\bar{M} = (\mathcal{S}, \mathcal{A}, \hat{r}, \bar{\mathsf{P}})$ as follows. For $s \in \mathcal{S}^0$, $\bar{\mathsf{P}}_{s,x}^a = \mathbf{1}_{x=s}$; And for $s \notin \mathcal{S}^0$, set $\bar{\mathsf{P}}_{s,x}^a = \mathsf{P}_{s,x}^a$.

Then for any $s \in \mathcal{S}^0$, from Lemma 3, it holds that $V_{\hat{\mathcal{P}}}^{\pi_r}(s) = 0$, and hence

$$V_{\hat{\mathcal{P}}}^{\pi_r}(s) - V_{\mathsf{P}}^{\pi_r}(s) \le 0. \tag{65}$$

It further implies that

$$V_{\hat{\mathcal{P}}}^{\pi_r}(s) - V_{\mathsf{P}}^{\pi_r}(s) = \bar{\mathsf{P}}_s^{\pi_r(s)}(V_{\hat{\mathcal{P}}}^{\pi_r} - V_{\mathsf{P}}^{\pi_r}) \le \gamma \bar{\mathsf{P}}_s^{\pi_r(s)}(V_{\hat{\mathcal{P}}}^{\pi_r} - V_{\mathsf{P}}^{\pi_r}). \tag{66}$$

On the other hand, for $s \notin \mathcal{S}^0$, Lemma 3 implies that $N(s, \pi_r(s)) > 0$, hence equation 32 implies $N(s, \pi_r(s)) > 0$, $\hat{r}(s, \pi_r(s)) = r(s, \pi_r(s))$. Hence

$$
\begin{aligned}
V_{\hat{\mathcal{P}}}^{\pi_r}(s) - V_{\mathsf{P}}^{\pi_r}(s) &\overset{(a)}{=} \gamma \sigma_{\hat{\mathcal{P}}_s^{\pi_r(s)}}(V_{\hat{\mathcal{P}}}^{\pi_r}) - \gamma \mathsf{P}_s^{\pi_r(s)} V_{\mathsf{P}}^{\pi_r} \\
&= \gamma (\sigma_{\hat{\mathcal{P}}_s^{\pi_r(s)}}(V_{\hat{\mathcal{P}}}^{\pi_r}) - \mathsf{P}_s^{\pi_r(s)} V_{\hat{\mathcal{P}}}^{\pi_r} + \mathsf{P}_s^{\pi_r(s)} V_{\hat{\mathcal{P}}}^{\pi_r} - \mathsf{P}_s^{\pi_r(s)} V_{\mathsf{P}}^{\pi_r}) \\
&= \gamma \mathsf{P}_s^{\pi_r(s)}(V_{\hat{\mathcal{P}}}^{\pi_r} - V_{\mathsf{P}}^{\pi_r}) + \gamma (\sigma_{\hat{\mathcal{P}}_s^{\pi_r(s)}}(V_{\hat{\mathcal{P}}}^{\pi_r}) - \mathsf{P}_s^{\pi_r(s)} V_{\hat{\mathcal{P}}}^{\pi_r}) \\
&\triangleq \gamma \bar{\mathsf{P}}_s^{\pi_r(s)}(V_{\hat{\mathcal{P}}}^{\pi_r} - V_{\mathsf{P}}^{\pi_r}) + c(s),
\end{aligned} \tag{67}
$$

where $(a)$ is from $r(s, \pi_r(s)) = \hat{r}(s, \pi_r(s))$, and $c(s) \triangleq \gamma(\sigma_{\hat{\mathcal{P}}_s^{\pi_r(s)}}(V_{\hat{\mathcal{P}}}^{\pi_r}) - \mathsf{P}_s^{\pi_r(s)} V_{\hat{\mathcal{P}}}^{\pi_r})$.

According to the bound we obtained in Lemma 8, it holds that

$$c(s) \le 2\sqrt{\frac{48 \log \frac{4SAN}{(1-\gamma)\delta} \epsilon_1}{(1-\gamma)N(s,\pi_r(s))}} + 2\epsilon_1 \sqrt{\frac{48 \log \frac{4SAN}{(1-\gamma)\delta}}{N(s,\pi_r(s))}} + \frac{48 \log \frac{4SAN}{(1-\gamma)\delta}}{(1-\gamma)N(s,\pi_r(s))}. \tag{68}$$

Combine eq. (66) and eq. (67), then

$$V_{\hat{\mathcal{P}}}^{\pi_r}(s) - V_{\mathsf{P}}^{\pi_r}(s) \le \gamma \bar{\mathsf{P}}_s^{\pi_r(s)}(V_{\hat{\mathcal{P}}}^{\pi_r} - V_{\mathsf{P}}^{\pi_r}) + \tilde{c}(s), \tag{69}$$

where

$$\tilde{c}(s) = \begin{cases} 2\sqrt{\frac{48 \log \frac{4SAN}{(1-\gamma)\delta} \epsilon_1}{(1-\gamma)N(s,\pi_r(s))}} + 2\epsilon_1 \sqrt{\frac{48 \log \frac{4SAN}{(1-\gamma)\delta}}{N(s,\pi_r(s))}} + \frac{48 \log \frac{4SAN}{(1-\gamma)\delta}}{(1-\gamma)N(s,\pi_r(s))}, & s \notin \mathcal{S}^0 \\ 0, & s \in \mathcal{S}^0 \end{cases} \tag{70}$$

Applying eq. (69) recursively further implies

$$\rho^\top \Delta_2 \le \frac{1}{1-\gamma} \langle \bar{d}^{\pi_r}, \tilde{c} \rangle, \tag{71}$$

where $\bar{d}^{\pi_r}$ is the discounted visitation distribution induced by $\pi_r$ and $\bar{\mathsf{P}}$.

Note that eq. (31) and Lemma 8 of (Shi & Chi, 2022) state that with probability $1 - \delta$, for any $(s,a)$ pair,

$$N(s,a) \ge \frac{N\mu(s,a)}{8 \log \frac{4SA}{\delta}}. \tag{72}$$

Hence under this event, $\tilde{c}(s)$ can be bounded as

$$\tilde{c}(s) \le 2\sqrt{\frac{384 \log^2 \frac{4SAN}{(1-\gamma)\delta} \epsilon_1}{(1-\gamma)N\mu_{\min}}} + 2\epsilon_1 \sqrt{\frac{384 \log^2 \frac{4SAN}{(1-\gamma)\delta}}{N\mu_{\min}}} + \frac{384 \log^2 \frac{4SAN}{(1-\gamma)\delta}}{(1-\gamma)N\mu_{\min}}. \tag{73}$$

Hence we have that

$$\rho^\top \Delta_2 \leq \frac{1}{1-\gamma} \langle \bar{d}^{\pi_r}, \tilde{c} \rangle$$

$$\leq 2\sqrt{\frac{384 \log^2 \frac{4SAN}{(1-\gamma)\delta} \epsilon_1}{(1-\gamma)^3 N \mu_{\min}}} + 2\epsilon_1 \sqrt{\frac{384 \log^2 \frac{4SAN}{(1-\gamma)\delta}}{(1-\gamma)^3 N \mu_{\min}}} + \frac{384 \log^2 \frac{4SAN}{(1-\gamma)\delta}}{(1-\gamma)^2 N \mu_{\min}}$$

$$\leq 2\sqrt{\frac{384 \log^2 \frac{4SAN}{(1-\gamma)\delta}}{(1-\gamma)^3 N^2 \mu_{\min}}} + 2\sqrt{\frac{384 \log^2 \frac{4SAN}{(1-\gamma)\delta}}{(1-\gamma)^3 N^3 \mu_{\min}}} + \frac{384 \log^2 \frac{4SAN}{(1-\gamma)\delta}}{(1-\gamma)^2 N \mu_{\min}}. \tag{74}$$

$\square$

## C.1 Auxiliary Lemmas

**Lemma 3.** *Recall the set* $\mathcal{S}^0 \triangleq \{s \in \mathcal{S} : N(s) = 0\}$*. Then*

*(1). For any policy* $\pi$ *and* $s \in \mathcal{S}^0$*,* $V_{\hat{\mathcal{P}}}^\pi(s) = 0$*;*

*(2). There exists a deterministic robust optimal policy* $\pi_r$*, such that for any* $s \notin \mathcal{S}^0$*,* $N(s, \pi_r(s)) > 0$*.*

*Proof.* **Proof of (1).**

For any $s \in \mathcal{S}^0$, it holds that $N(s, a) = 0$ for any $a \in \mathcal{A}$. Hence $\hat{r}(s, a) = 0$ and $\hat{\mathcal{P}}_s^a = \Delta(\mathcal{S})$.

Then for any policy $\pi$ and $a \in \mathcal{A}$, it holds that

$$Q_{\hat{\mathcal{P}}}^\pi(s, a) = \hat{r}(s, a) + \gamma \sigma_{\hat{\mathcal{P}}_s^a}(V_{\hat{\mathcal{P}}}^\pi) \leq \gamma V_{\hat{\mathcal{P}}}^\pi(s). \tag{75}$$

Thus

$$V_{\hat{\mathcal{P}}}^\pi(s) = \sum_a \pi(a|s) Q_{\hat{\mathcal{P}}}^\pi(s, a) \leq \gamma V_{\hat{\mathcal{P}}}^\pi(s), \tag{76}$$

which implies $V_{\hat{\mathcal{P}}}^\pi(s) = 0$ together with $V_{\hat{\mathcal{P}}}^\pi \geq 0$.

**Proof of (2).**

We prove Claim (2) by contradiction. Assume that for any optimal policy $\pi_r$, there exists $s \notin \mathcal{S}^0$ such that $N(s, \pi_r(s)) = 0$. We then consider a fixed pair $(\pi_r, s)$.

$N(s, \pi_r(s)) = 0$ further implies $\hat{r}(s, \pi_r(s)) = 0$, $\hat{\mathcal{P}}_s^{\pi_r(s)} = \Delta(\mathcal{S})$, and

$$V_{\hat{\mathcal{P}}}^{\pi_r}(s) = \max_a Q_{\hat{\mathcal{P}}}^{\pi_r}(s, a) = Q_{\hat{\mathcal{P}}}^{\pi_r}(s, \pi_r(s)) = \hat{r}(s, \pi_r(s)) + \gamma \sigma_{\hat{\mathcal{P}}_s^{\pi_r(s)}}(V_{\hat{\mathcal{P}}}^{\pi_r}) \leq \gamma V_{\hat{\mathcal{P}}}^{\pi_r}(s), \tag{77}$$

where the last inequality is from $\hat{\mathcal{P}}_s^{\pi_r(s)} = \Delta(\mathcal{S})$, $\hat{r}(s, \pi_r(s)) = 0$, and $\sigma_{\hat{\mathcal{P}}_s^{\pi_r(s)}}(V_{\hat{\mathcal{P}}}^{\pi_r}) \leq \mathbf{1}_s V_{\hat{\mathcal{P}}}^{\pi_r} = V_{\hat{\mathcal{P}}}^{\pi_r}(s)$. This further implies that $V_{\hat{\mathcal{P}}}^{\pi_r}(s) = 0$ because $V_{\hat{\mathcal{P}}}^{\pi_r} \geq 0$.

On the other hand, since $s \notin \mathcal{S}^0$, there exists another action $b \neq \pi_r(s)$ such that $N(s, b) > 0$, and hence $\hat{r}(s, b) = r(s, b)$. We consider the following two cases.

(I). If $r(s, b) > 0$, then

$$Q_{\hat{\mathcal{P}}}^{\pi_r}(s, b) = \hat{r}(s, b) + \gamma \sigma_{\hat{\mathcal{P}}_s^b}(V_{\hat{\mathcal{P}}}^{\pi_r}) > 0 = Q_{\hat{\mathcal{P}}}^{\pi_r}(s, \pi_r(s)), \tag{78}$$

which is contradict to $V_{\hat{\mathcal{P}}}^{\pi_r}(s) = \max_a Q_{\hat{\mathcal{P}}}^{\pi_r}(s, a) = Q_{\hat{\mathcal{P}}}^{\pi_r}(s, \pi_r(s))$.

(II). If $r(s, b) = 0$, Lemma 4 then implies the modified policy $f_b^s(\pi_r)$ is also optimal, and satisfies $N(x, f_b^s(\pi_r)(x)) = N(x, \pi_r(x))$ for any $x \neq s$, and $N(s, f_b^s(\pi_r)(s)) > 0$.

Then consider the modified policy $f_b^s(\pi_r)$.

If there still exists $s' \notin \mathcal{S}^0$ such that $N(s', f_b^s(\pi_r)(s')) = 0$, then similarly, there exists another action $b' \neq f_b^s(\pi_r)(s')$ such that $N(s', b') > 0$. Then whether $r(s', b') > 0$, which falls into Case (I) and leads to a contradiction, or applying Lemma 4 again implies another optimal policy $f_{b'}^{s'}(f_b^s(\pi_r))$,

such that $N(s, f_{b'}^{s'}(f_b^s(\pi_r))(x)) = N(s, f_b^s(\pi_r)(x)) > 0$ for $x \notin \{s, s'\}$, $N(s, f_{b'}^{s'}(f_b^s(\pi_r))(s)) = N(s, f_b^s(\pi_r)(s)) > 0$ and $N(s', f_{b'}^{s'}(f_b^s(\pi_r))(s')) > 0$.

Repeating this procedure recursively further implies there exists an optimal policy $\pi$, such that $N(s, \pi(s)) > 0$ for any $s \notin \mathcal{S}^0$, which is a contraction to our assumption.

Therefore it completes the proof. □

**Lemma 4.** *For a robust optimal policy $\pi_r$, if there exists a state $s \notin \mathcal{S}^0$ and an action $b$ such that $N(s, \pi_r(s)) = 0$, $r(s, b) = 0$ and $N(s, b) > 0$, define a modified policy $f_b^s(\pi_r)$ as*

$$f_b^s(\pi_r)(s) = b, \tag{79}$$
$$f_b^s(\pi_r)(x) = \pi_r(x), \text{ for } x \neq s. \tag{80}$$

*Then the modified policy $f_b^s(\pi_r)$ is also optimal, and satisfies $N(s, f_b^s(\pi_r)(s)) > 0$, $N(x, f_b^s(\pi_r)(x)) = N(x, \pi_r(x)), \forall x \neq s$.*

*Proof.* Recall that $\hat{\mathcal{P}}_s^{\pi_r(s)} = \Delta(\mathcal{S})$ and $\hat{\mathcal{P}}_s^b \subset \Delta(\mathcal{S})$, we have that

$$V_{\hat{\mathcal{P}}}^{f_b^s(\pi_r)} \geq V_{\hat{\mathcal{P}}_s^b}^{f_b^s(\pi_r)}, \tag{81}$$

where $\hat{\mathcal{P}}_s^b$ is a modified uncertainty set defined as

$$(\hat{\mathcal{P}}_s^b)_s^b = \Delta(\mathcal{S}), \tag{82}$$
$$(\hat{\mathcal{P}}_s^b)_x^a = \hat{\mathcal{P}}_x^a, \text{ for } (x, a) \neq (s, b). \tag{83}$$

Now we have that

$$V_{\hat{\mathcal{P}}_s^b}^{f_b^s(\pi_r)}(s) = Q_{\hat{\mathcal{P}}_s^b}^{f_b^s(\pi_r)}(s, b) = r(s, b) + \gamma\sigma_{(\hat{\mathcal{P}}_s^b)_s^b}(V_{\hat{\mathcal{P}}_s^b}^{f_b^s(\pi_r)}) \leq \gamma V_{\hat{\mathcal{P}}_s^b}^{f_b^s(\pi_r)}(s), \tag{84}$$

which further implies $V_{\hat{\mathcal{P}}_s^b}^{f_b^s(\pi_r)}(s) = 0$. Note that in eq. (77), we have shown $V_{\hat{\mathcal{P}}}^{\pi_r}(s) = 0$, hence $V_{\hat{\mathcal{P}}_s^b}^{f_b^s(\pi_r)}(s) = V_{\hat{\mathcal{P}}}^{\pi_r}(s) = 0$.

Now consider the two robust Bellman operator $\mathbf{T}_b^s V(x) = \sum_a f_b^s(\pi_r)(a|x)(\hat{r}(x, a) + \gamma\sigma_{(\hat{\mathcal{P}}_s^b)_x^a}(V))$ and $\mathbf{T}V(x) = \hat{r}(x, \pi_r(x)) + \gamma\sigma_{\hat{\mathcal{P}}_x^{\pi_r(x)}}(V)$. It is known that $V_{\hat{\mathcal{P}}_s^b}^{f_b^s(\pi_r)}$ is the unique fixed point of the robust Bellman operator $\mathbf{T}_b^s$ and $V_{\hat{\mathcal{P}}}^{\pi_r}$ is the unique fixed point of $\mathbf{T}$.

When $x \neq s$,

$$\begin{aligned}
\mathbf{T}_b^s V_{\hat{\mathcal{P}}}^{\pi_r}(x) &= \sum_a f_b^s(\pi_r)(a|x)(\hat{r}(x, a) + \gamma\sigma_{(\hat{\mathcal{P}}_s^b)_x^a}(V_{\hat{\mathcal{P}}}^{\pi_r})) \\
&\stackrel{(a)}{=} \sum_a \pi_r(a|x)(\hat{r}(x, a) + \gamma\sigma_{(\hat{\mathcal{P}}_s^b)_x^a}(V_{\hat{\mathcal{P}}}^{\pi_r})) \\
&= \hat{r}(x, \pi_r(x)) + \gamma\sigma_{(\hat{\mathcal{P}}_s^b)_x^{\pi_r(x)}}(V_{\hat{\mathcal{P}}}^{\pi_r}) \\
&\stackrel{(b)}{=} \hat{r}(x, \pi_r(x)) + \gamma\sigma_{\hat{\mathcal{P}}_x^{\pi_r(x)}}(V_{\hat{\mathcal{P}}}^{\pi_r}) \\
&= \mathbf{T}V_{\hat{\mathcal{P}}}^{\pi_r}(x) = V_{\hat{\mathcal{P}}}^{\pi_r}(x),
\end{aligned} \tag{85}$$

where $(a)$ is from $f_b^s(\pi_r)(x) = \pi_r(x)$ when $x \neq s$, $(b)$ is from $(\hat{\mathcal{P}}_s^b)_x^{\pi_r(x)} = \hat{\mathcal{P}}_x^{\pi_r(x)}$.

And for $s$, it holds that

$$\begin{aligned}
\mathbf{T}_b^s V_{\hat{\mathcal{P}}}^{\pi_r}(s) &= \hat{r}(s, b) + \gamma\sigma_{(\hat{\mathcal{P}}_s^b)_s^b}(V_{\hat{\mathcal{P}}}^{\pi_r}) \\
&\stackrel{(a)}{=} \hat{r}(s, \pi_r(s)) + \gamma\sigma_{\Delta(\mathcal{S})}(V_{\hat{\mathcal{P}}}^{\pi_r}) \\
&\stackrel{(b)}{=} \hat{r}(s, \pi_r(s)) + \gamma\sigma_{\hat{\mathcal{P}}_s^{\pi_r(s)}}(V_{\hat{\mathcal{P}}}^{\pi_r})
\end{aligned}$$

$$
\begin{aligned}
&= \mathbf{T}V_{\hat{\mathcal{P}}}^{\pi_r}(s) \\
&= V_{\hat{\mathcal{P}}}^{\pi_r}(s),
\end{aligned}
\tag{86}
$$

where $(a)$ is from $(\hat{\mathcal{P}}_s^b)_s^b = \Delta(\mathcal{S})$ and $\hat{r}(s, b) = r(s, b) = 0 = \hat{r}(s, \pi_r(s))$, and $(b)$ follows from the fact $\hat{\mathcal{P}}_s^{\pi_r(s)} = \Delta(\mathcal{S})$.

eq. (85) and eq. (86) further imply that $V_{\hat{\mathcal{P}}}^{\pi_r}$ is also a fixed point of $\mathbf{T}_b^s$. Hence it must be identical to $V_{\hat{\mathcal{P}}_s^b}^{f_b^s(\pi_r)}$, i.e., $V_{\hat{\mathcal{P}}_s^b}^{f_b^s(\pi_r)} = V_{\hat{\mathcal{P}}}^{\pi_r}$.

Combine with eq. (81), we have

$$
V_{\hat{\mathcal{P}}}^{f_b^s(\pi_r)} \geq V_{\hat{\mathcal{P}}_s^b}^{f_b^s(\pi_r)} = V_{\hat{\mathcal{P}}}^{\pi_r},
\tag{87}
$$

which implies that $f_b^s(\pi_r)$ is also optimal with $N(s, f_b^s(\pi_r)(s)) = N(s, b) > 0$. And since $f_b^s(\pi_r)(x) = \pi_r(x)$ for $x \neq s$, then $N(x, f_b^s(\pi_r)(x)) = N(x, \pi_r(x))$. This thus completes the proof. □

**Lemma 5** (Lemma 4, (Li et al., 2022)). *For any $\delta$, with probability $1 - \delta$, $\max\{12N(s, a), 8\log \frac{NS}{\delta}\} \geq N\mu(s, a), \forall s, a$.*

**Lemma 6** (Lemma 9, (Li et al., 2022)). *For any $(s, a)$ pair with $N(s, a) > 0$, if $V$ is an vector independent of $\hat{\mathsf{P}}_s^a$ obeying $\|V\| \leq \frac{1}{1-\gamma}$, then with probability at least $1 - \delta$,*

$$
|(\hat{\mathsf{P}}_s^a - \mathsf{P}_s^a)V| \leq \sqrt{\frac{48 \mathbf{Var}_{\hat{\mathsf{P}}_s^a}(V)\log\frac{4N}{\delta}}{N(s, a)}} + \frac{48\log\frac{4N}{\delta}}{(1-\gamma)N(s, a)},
\tag{88}
$$

$$
\mathbf{Var}_{\hat{\mathsf{P}}_s^a}(V) \leq 2\mathbf{Var}_{\mathsf{P}_s^a}(V) + \frac{5\log\frac{4N}{\delta}}{3(1-\gamma)^2 N(s, a)}.
\tag{89}
$$

**Lemma 7.** *Suppose $\gamma \in [0.5, 1)$, with probability at least $1 - \delta$, it holds*

$$
|(\hat{\mathsf{P}}_s^a - \mathsf{P}_s^a)V_{\hat{\mathcal{P}}}^{\pi_r}| \leq 12\sqrt{\frac{\mathbf{Var}_{\hat{\mathsf{P}}_s^a}(V_{\hat{\mathcal{P}}}^{\pi_r})\log\frac{4N}{\delta}}{N(s, a)}} + \frac{74\log\frac{4N}{\delta}}{(1-\gamma)N(s, a)},
\tag{90}
$$

$$
\mathbf{Var}_{\hat{\mathsf{P}}_s^a}(V_{\hat{\mathcal{P}}}^{\pi_r}) \leq 2\mathbf{Var}_{\mathsf{P}_s^a}(V_{\hat{\mathcal{P}}}^{\pi_r}) + \frac{41\log\frac{4N}{\delta}}{(1-\gamma)^2 N(s, a)}.
\tag{91}
$$

*simultaneously for any pair $(s, a) \in \mathcal{S} \times \mathcal{A}$.*

*Proof.* When $N(s, a) = 0$, the results hold naturally. We hence only consider $(s, a)$ with $N(s, a) > 0$.
**Part 1**.

Recall that $\hat{\mathsf{M}} = (\mathcal{S}, \mathcal{A}, \hat{\mathsf{P}}, \hat{r})$ is the estimated MDP. For any state $s$ and positive scalar $u > 0$, we first construct an auxiliary state-absorbing MDP $\hat{\mathsf{M}}^{s,u} = (\mathcal{S}, \mathcal{A}, \mathsf{P}^{s,u}, r^{s,u})$ as follows.

For all states except $s$, the MDP structure of $\hat{\mathsf{M}}^{s,u}$ is identical to $\hat{\mathsf{M}}$, i.e., for any $x \neq s$ and $a \in \mathcal{A}$,

$$
(\mathsf{P}^{s,u})_{x,\cdot}^a = \hat{\mathsf{P}}_{x,\cdot}^a, r^{s,u}(x, a) = \hat{r}(x, a);
\tag{92}
$$

State $s$ is an absorbing state in $\hat{\mathsf{M}}^{s,u}$, namely, for any $a \in \mathcal{A}$,

$$
(\mathsf{P}^{s,u})_{s,x}^a = \mathbf{1}_{x=s}, r^{s,u}(s, a) = u.
\tag{93}
$$

We then define a robust MDP $\mathcal{M}^{s,u} = (\mathcal{S}, \mathcal{A}, \mathcal{P}^{s,u}, r^{s,u})$ centered at $\hat{\mathsf{M}}^{s,u}$ as following: the uncertainty set $\mathcal{P}^{s,u}$ is defined as $\hat{\mathcal{P}} = \bigotimes_{x,a}(\mathcal{P}^{s,u})_x^a$, where if $x \neq s$,

$$
(\mathcal{P}^{s,u})_x^a = \left\{ q \in \Delta(\mathcal{S}) : \|q - (\mathsf{P}^{s,u})_x^a\| \leq \min\left\{2, \frac{\log\frac{N}{\delta}}{N(x, a)}\right\} \right\}
\tag{94}
$$

and

$$(\mathcal{P}^{s,u})^a_s = \{\mathbf{1}_s\}. \tag{95}$$

The optimal robust value function of $\mathcal{M}^{s,u}$ is denoted by $V^{s,u}$.

**Part 2**. We claim that if we choose $u^* = (1-\gamma)V^{\pi_r}_{\hat{\mathcal{P}}}(s)$, then $V^{s,u^*} = V^{\pi_r}_{\hat{\mathcal{P}}}$. We prove this as follows.

Firstly note that the function $V^{s,u^*}$ is the unique fixed point of the operator $\mathbf{T}^{s,u^*}(V)(x) = \max_a\{r^{s,u^*}(x,a) + \gamma\sigma_{(\mathcal{P}^{s,u^*})^a_x}(V)\}$.

For $x \neq s$, we note that

$$\begin{aligned}
\mathbf{T}^{s,u^*}(V^{\pi_r}_{\hat{\mathcal{P}}})(x) &= \max_a\{r^{s,u^*}(x,a) + \gamma\sigma_{(\mathcal{P}^{s,u^*})^a_x}(V^{\pi_r}_{\hat{\mathcal{P}}})\} \\
&= \max_a\{\hat{r}(x,a) + \gamma\sigma_{\hat{\mathcal{P}}^a_x}(V^{\pi_r}_{\hat{\mathcal{P}}})\} \\
&= V^{\pi_r}_{\hat{\mathcal{P}}}(x), \tag{96}
\end{aligned}$$

which is because $r^{s,u^*}(x,a) = \hat{r}(x,a)$ and $(\mathcal{P}^{s,u^*})^a_x = \hat{\mathcal{P}}^a_x$ for $x \neq s$.

For $s$, we have that

$$\begin{aligned}
\mathbf{T}^{s,u^*}(V^{\pi_r}_{\hat{\mathcal{P}}})(s) &= \max_a\{r^{s,u^*}(s,a) + \gamma\sigma_{(\mathcal{P}^{s,u^*})^a_s}(V^{\pi_r}_{\hat{\mathcal{P}}})\} \\
&= \max_a\{u^* + \gamma\sigma_{(\mathcal{P}^{s,u^*})^a_s}(V^{\pi_r}_{\hat{\mathcal{P}}})\} \\
&= \max_a\{(1-\gamma)V^{\pi_r}_{\hat{\mathcal{P}}}(s) + \gamma(V^{\pi_r}_{\hat{\mathcal{P}}})(s)\} \\
&= V^{\pi_r}_{\hat{\mathcal{P}}}(s), \tag{97}
\end{aligned}$$

which from $(\mathcal{P}^{s,u^*})^a_s = \{\mathbf{1}_s\}$.

Hence combining with eq. (96) implies that $V^{\pi_r}_{\hat{\mathcal{P}}}$ is also a fixed point of $\mathbf{T}^{s,u^*}$, and hence it must be identical to $V^{s,u^*}$, which proves our claim.

**Part 3**. Define a set $U_c \triangleq \{\frac{i}{N}|i = 1, ..., N\}$. Clearly, $U_c$ is a $\frac{1}{N}$-net (Vershynin, 2018; Li et al., 2022) of the interval $[0,1]$.

Note that for any $u \in U_c$, $\mathcal{P}^{s,u}$ is independent with $\hat{\mathsf{P}}^a_s$, hence $V^{s,u}$ is also independent with $\hat{\mathsf{P}}^a_s$. Also, since $u \leq 1$, $\|V^{s,u}\| \leq \frac{1}{1-\gamma}$.

Then invoking Lemma 6 implies that for any $N(s,a) > 0$, with probability at least $1 - \delta$, it holds simultaneously for all $u \in U_c$ that

$$|(\hat{\mathsf{P}}^a_s - \mathsf{P}^a_s)V^{s,u}| \leq \sqrt{\frac{48\mathbf{Var}_{\hat{\mathsf{P}}^a_s}(V^{s,u})\log\frac{4N^2}{\delta}}{N(s,a)}} + \frac{48\log\frac{4N^2}{\delta}}{(1-\gamma)N(s,a)}, \tag{98}$$

$$\mathbf{Var}_{\hat{\mathsf{P}}^a_s}(V^{s,u}) \leq 2\mathbf{Var}_{\mathsf{P}^a_s}(V^{s,u}) + \frac{5\log\frac{4N^2}{\delta}}{3(1-\gamma)^2 N(s,a)}. \tag{99}$$

**Part 4**. Since $u^* = (1-\gamma)V^{\pi_r}_{\hat{\mathcal{P}}} \leq 1$, then there exists $u_0 \in U_c$, such that $|u_0 - u^*| \leq \frac{1}{N}$. Moreover, we claim that

$$\|V^{s,u^*} - V^{s,u_0}\| \leq \frac{1}{N(1-\gamma)}. \tag{100}$$

To prove eq. (100), first note that

$$\begin{aligned}
|V^{s,u^*}(s) - V^{s,u_0}(s)| &\leq \max_a|(u^* - u_0) + \gamma(\sigma_{(\mathcal{P}^{s,u^*})^a_s}(V^{s,u^*}) - \sigma_{(\mathcal{P}^{s,u_0})^a_s}(V^{s,u_0}))| \\
&\overset{(a)}{\leq} |u^* - u_0| + \gamma\max_a|\sigma_{(\mathcal{P}^{s,u^*})^a_s}(V^{s,u^*}) - \sigma_{(\mathcal{P}^{s,u^*})^a_s}(V^{s,u_0})| \\
&\overset{(b)}{\leq} |u^* - u_0| + \gamma\|V^{s,u^*} - V^{s,u_0}\|, \tag{101}
\end{aligned}$$

where $(a)$ is because $(\mathcal{P}^{s,u_0})^a_s = (\mathcal{P}^{s,u^*})^a_s = \{\mathbf{1}_s\}$, and $(b)$ is due to the non-expansion of the support function (Lemma 1, (Panaganti & Kalathil, 2022)).

For $x \neq s$, we have that

$$|V^{s,u^*}(x) - V^{s,u_0}(x)| \leq \max_a |\hat{r}(x,a) - \hat{r}(x,a) + \gamma(\sigma_{\hat{\mathcal{P}}^a_x}(V^{s,u^*}) - \sigma_{\hat{\mathcal{P}}^a_x}(V^{s,u_0}))|$$

$$\leq \gamma \|V^{s,u^*} - V^{s,u_0}\|. \tag{102}$$

Thus by combining eq. (101) and eq. (102), we have that

$$\|V^{s,u^*} - V^{s,u_0}\| \leq \frac{1}{N} + \gamma \|V^{s,u^*} - V^{s,u_0}\|, \tag{103}$$

and hence proof the claim eq. (100).

Therefore,

$$\mathbf{Var}_{\mathsf{P}^a_s}(V^{s,u_0}) - \mathbf{Var}_{\mathsf{P}^a_s}(V^{s,u^*})$$

$$= \mathsf{P}^a_s \left( (V^{s,u_0} - \mathsf{P}^a_s V^{s,u_0}) \circ (V^{s,u_0} - \mathsf{P}^a_s V^{s,u_0}) - (V^{s,u^*} - \mathsf{P}^a_s V^{s,u^*}) \circ (V^{s,u^*} - \mathsf{P}^a_s V^{s,u^*}) \right)$$

$$\overset{(a)}{\leq} \mathsf{P}^a_s \left( (V^{s,u_0} - \mathsf{P}^a_s V^{s,u^*}) \circ (V^{s,u_0} - \mathsf{P}^a_s V^{s,u^*}) - (V^{s,u^*} - \mathsf{P}^a_s V^{s,u^*}) \circ (V^{s,u^*} - \mathsf{P}^a_s V^{s,u^*}) \right)$$

$$\leq \mathsf{P}^a_s \left( (V^{s,u_0} - \mathsf{P}^a_s V^{s,u^*} + V^{s,u^*} - \mathsf{P}^a_s V^{s,u^*}) \circ (V^{s,u_0} - V^{s,u^*}) \right)$$

$$\leq \frac{2}{1-\gamma} |\mathsf{P}^a_s (V^{s,u_0} - V^{s,u^*})|$$

$$\leq \frac{2}{N(1-\gamma)^2}, \tag{104}$$

where $(a)$ is due to the fact $\mathbb{E}[X] = \arg\min_c \mathbb{E}[(X-c)^2]$, and the last inequality is due to $\|V^{s,u_0}\| \leq \frac{1}{1-\gamma}, \|V^{s,u^*}\| \leq \frac{1}{1-\gamma}$.

Similarly, swapping $V^{s,u_0}$ and $V^{s,u^*}$ implies

$$\mathbf{Var}_{\mathsf{P}^a_s}(V^{s,u^*}) - \mathbf{Var}_{\mathsf{P}^a_s}(V^{s,u_0}) \leq \frac{2}{N(1-\gamma)^2}, \tag{105}$$

and further

$$|\mathbf{Var}_{\mathsf{P}^a_s}(V^{s,u^*}) - \mathbf{Var}_{\mathsf{P}^a_s}(V^{s,u_0})| \leq \frac{2}{N(1-\gamma)^2}. \tag{106}$$

We note that eq. (106) is exactly identical to (159) in Section A.4 of (Li et al., 2022), and hence the remaining proof can be obtained by following the proof in Section A.4 in (Li et al., 2022), and are omitted here. $\square$

**Lemma 8.** *With probability at least $1 - \delta$, it holds that for any $s, a$,*

$$\hat{\mathsf{P}}^a_s V^{\pi_r}_{\hat{\mathcal{P}}} - \mathsf{P}^a_s V^{\pi_r}_{\hat{\mathcal{P}}}$$

$$\leq \hat{\mathsf{P}}^a_s V^{\pi_r}_{\hat{\mathcal{P}}} - \sigma_{\hat{\mathcal{P}}^a_s}(V^{\pi_r}_{\hat{\mathcal{P}}}) + 2\sqrt{\frac{48\log\frac{4SAN}{(1-\gamma)\delta}\epsilon_1}{(1-\gamma)N(s,a)}} + 2\epsilon_1\sqrt{\frac{48\log\frac{4SAN}{(1-\gamma)\delta}}{N(s,a)}} + \frac{96\log\frac{4SAN}{(1-\gamma)\delta}}{(1-\gamma)N(s,a)}. \tag{107}$$

*Proof.* We first show the inequality above holds for any $V \in \left[0, \frac{1}{1-\gamma}\right]$ that is independent with $\hat{\mathsf{P}}^a_s$.

From the duality form of the $\sigma_{\mathcal{P}}(V)$ (Iyengar, 2005), it holds that

$$\hat{\mathsf{P}}^a_s V - \sigma_{\hat{\mathcal{P}}^a_s}(V) = \hat{\mathsf{P}}^a_s V - \max_{\alpha \in [V_{\min}, V_{\max}]} \left\{ \hat{\mathsf{P}}^a_s V_\alpha - \sqrt{R^a_s \mathbf{Var}_{\hat{\mathsf{P}}^a_s}(V_\alpha)} \right\}$$

$$= \min_{\alpha \in [V_{\min}, V_{\max}]} \left\{ \hat{\mathsf{P}}^a_s(V - V_\alpha) + \sqrt{R^a_s \mathbf{Var}_{\hat{\mathsf{P}}^a_s}(V_\alpha)} \right\}, \tag{108}$$

where $V_\alpha \in \mathbb{R}^S$ and $V_\alpha(s) = \min\{V(s), \alpha\}$.

We denote the optimum of the optimization by $\alpha^*$, i.e.,

$$\hat{\mathsf{P}}_s^a(V - V_{\alpha^*}) + \sqrt{R_s^a \mathbf{Var}_{\hat{\mathsf{P}}_s^a}(V_{\alpha^*})} = \min_{\alpha \in [V_{\min}, V_{\max}]} \left\{ \hat{\mathsf{P}}_s^a(V - V_\alpha) + \sqrt{R_s^a \mathbf{Var}_{\hat{\mathsf{P}}_s^a}(V_\alpha)} \right\}. \quad (109)$$

Then eq. (108) can be further bounded as

$$\hat{\mathsf{P}}_s^a V - \sigma_{\hat{\mathcal{P}}_s^a}(V) = \hat{\mathsf{P}}_s^a(V - V_{\alpha^*}) + \sqrt{R_s^a \mathbf{Var}_{\hat{\mathsf{P}}_s^a}(V_{\alpha^*})}$$

$$\geq \hat{\mathsf{P}}_s^a(V - V_{\alpha^*}) - \mathsf{P}_s^a(V - V_{\alpha^*}) + \sqrt{R_s^a \mathbf{Var}_{\hat{\mathsf{P}}_s^a}(V_{\alpha^*})}, \quad (110)$$

where the inequality is due to the fact that $V(s) \geq V_\alpha(s)$.

On the other hand, for any $\alpha \in [V_{\min}, V_{\max}]$ that is fixed and independent with $\hat{\mathsf{P}}_s^a$, we have that

$$\hat{\mathsf{P}}_s^a V - \mathsf{P}_s^a V = \hat{\mathsf{P}}_s^a V_\alpha - \mathsf{P}_s^a V_\alpha + \hat{\mathsf{P}}_s^a(V - V_\alpha) - \mathsf{P}_s^a(V - V_\alpha)$$

$$\leq \hat{\mathsf{P}}_s^a(V - V_\alpha) - \mathsf{P}_s^a(V - V_\alpha) + \frac{48 \log \frac{4N}{\delta}}{(1-\gamma)N(s,a)} + \sqrt{\frac{48 \log \frac{4N}{\delta} \mathbf{Var}_{\hat{\mathsf{P}}_s^a}(V_\alpha)}{N(s,a)}}, \quad (111)$$

which is due to $\alpha$ is independent from $\hat{\mathsf{P}}_s^a$, and applying the Bernstein's inequality and (102) of (Shi et al., 2023). Moreover, it implies that

$$\hat{\mathsf{P}}_s^a V - \mathsf{P}_s^a V$$

$$\leq \hat{\mathsf{P}}_s^a(V - V_{\alpha^*}) - \mathsf{P}_s^a(V - V_{\alpha^*}) + \frac{48 \log \frac{4N}{\delta}}{(1-\gamma)N(s,a)} + \sqrt{\frac{48 \log \frac{4N}{\delta} \mathbf{Var}_{\hat{\mathsf{P}}_s^a}(V_{\alpha^*})}{N(s,a)}}$$

$$+ \left( \sqrt{\frac{48 \log \frac{4N}{\delta} \mathbf{Var}_{\hat{\mathsf{P}}_s^a}(V_\alpha)}{N(s,a)}} - \sqrt{\frac{48 \log \frac{4N}{\delta} \mathbf{Var}_{\hat{\mathsf{P}}_s^a}(V_{\alpha^*})}{N(s,a)}} \right) + (\hat{\mathsf{P}}_s^a(V_{\alpha^*} - V_\alpha) - \mathsf{P}_s^a(V_{\alpha^*} - V_\alpha)).$$

$$(112)$$

We now construct an $\epsilon_1$-Net (Vershynin, 2018) of $\left[0, \frac{1}{1-\gamma}\right]$ with $\epsilon_1 = \frac{1}{N}$. Specifically, there exists $\mathcal{U} = \left\{ \alpha_1, \alpha_2, ..., \alpha_m | \alpha_i \in \left[0, \frac{1}{1-\gamma}\right] \right\}$, such that for any $\alpha \in \left[0, \frac{1}{1-\gamma}\right]$, there exists $\alpha_j \in \mathcal{U}$ with $|\alpha - \alpha_j| \leq \epsilon_1$. Since $\alpha^* \in \left[0, \frac{1}{1-\gamma}\right]$, there exists $\beta \in \mathcal{U}$ with $|\beta - \alpha^*| \leq \epsilon_1$.

It is straightforward to see that

$$|V_{\alpha^*} - V_\beta| \leq |\beta - \alpha^*| \leq \epsilon_1, \quad (113)$$

and similarly following (207) of (Shi et al., 2023) implies that

$$\left| \sqrt{\mathbf{Var}_{\hat{\mathsf{P}}_s^a}(V_\beta)} - \sqrt{\mathbf{Var}_{\hat{\mathsf{P}}_s^a}(V_{\alpha^*})} \right| \leq 2\sqrt{\frac{\epsilon_1}{1-\gamma}}. \quad (114)$$

Hence we set $\alpha = \beta$ in eq. (112), take the union bound over $\mathcal{S}, \mathcal{A}$ and $\mathcal{U}$, and plug in the two inequalities above, we have that

$$\hat{\mathsf{P}}_s^a V - \mathsf{P}_s^a V$$

$$\leq \hat{\mathsf{P}}_s^a(V - V_{\alpha^*}) - \mathsf{P}_s^a(V - V_{\alpha^*}) + \frac{48 \log \frac{4SAN}{(1-\gamma)\delta}}{(1-\gamma)N(s,a)} + \sqrt{\frac{48 \log \frac{4SAN}{(1-\gamma)\delta} \mathbf{Var}_{\hat{\mathsf{P}}_s^a}(V_{\alpha^*})}{N(s,a)}}$$

$$+ 2\sqrt{\frac{48 \log \frac{4SAN}{(1-\gamma)\delta} \epsilon_1}{(1-\gamma)N(s,a)}} + 2\epsilon_1 \sqrt{\frac{48 \log \frac{4SAN}{(1-\gamma)\delta}}{N(s,a)}}. \quad (115)$$

Involving eq. (110) further implies that

$$\hat{\mathsf{P}}_s^a V - \mathsf{P}_s^a V$$

$$\leq \hat{\mathsf{P}}_s^a V - \sigma_{\hat{\mathcal{P}}_s^a}(V) + 2\sqrt{\frac{48 \log \frac{4SAN}{(1-\gamma)\delta}\epsilon_1}{(1-\gamma)N(s,a)}} + 2\epsilon_1 \sqrt{\frac{48 \log \frac{4SAN}{(1-\gamma)\delta}}{N(s,a)}} + \frac{48 \log \frac{4SAN}{(1-\gamma)\delta}}{(1-\gamma)N(s,a)}, \quad (116)$$

and hence

$$\sigma_{\hat{\mathcal{P}}_s^a}(V) - \mathsf{P}_s^a V \leq 2\sqrt{\frac{48 \log \frac{4SAN}{(1-\gamma)\delta}\epsilon_1}{(1-\gamma)N(s,a)}} + 2\epsilon_1 \sqrt{\frac{48 \log \frac{4SAN}{(1-\gamma)\delta}}{N(s,a)}} + \frac{48 \log \frac{4SAN}{(1-\gamma)\delta}}{(1-\gamma)N(s,a)}. \quad (117)$$

Now we consider $V_{\hat{\mathcal{P}}}^{\pi_r}$. Following the $\frac{1}{N}$-net $\mathcal{U}_2$ we constructed in Lemma 7, $V_{\hat{\mathcal{P}}}^{\pi_r} = V^{s,u}$ for some $u \in [0,1]$. Hence there exists some $u_i \in \mathcal{U}_2$ with $|u - u_i| \leq \frac{1}{N}$. Note that

$$
\begin{aligned}
&\sigma_{\hat{\mathcal{P}}_s^a}(V_{\hat{\mathcal{P}}}^{\pi_r}) - \mathsf{P}_s^a V_{\hat{\mathcal{P}}}^{\pi_r} \\
&= \sigma_{\hat{\mathcal{P}}_s^a}(V^{s,u}) - \mathsf{P}_s^a V^{s,u} \\
&= \sigma_{\hat{\mathcal{P}}_s^a}(V^{s,u_i}) - \mathsf{P}_s^a V^{s,u_i} \\
&\quad + \sigma_{\hat{\mathcal{P}}_s^a}(V^{s,u}) - \sigma_{\hat{\mathcal{P}}_s^a}(V^{s,u_i}) + \mathsf{P}_s^a V^{s,u_i} - \mathsf{P}_s^a V^{s,u} \\
&\leq 2\sqrt{\frac{48 \log \frac{4SAN}{(1-\gamma)\delta}\epsilon_1}{(1-\gamma)N(s,a)}} + 2\epsilon_1 \sqrt{\frac{48 \log \frac{4SAN}{(1-\gamma)\delta}}{N(s,a)}} + \frac{48 \log \frac{4SAN}{(1-\gamma)\delta}}{(1-\gamma)N(s,a)} \\
&\quad + \sigma_{\hat{\mathcal{P}}_s^a}(V^{s,u}) - \sigma_{\hat{\mathcal{P}}_s^a}(V^{s,u_i}) + \mathsf{P}_s^a V^{s,u_i} - \mathsf{P}_s^a V^{s,u}, \quad (118)
\end{aligned}
$$

which is due to $V^{s,u}$ is independent with $\hat{\mathsf{P}}_s^a$. Moreover, note that both $\sigma_{\hat{\mathcal{P}}_s^a}(V)$ and $\hat{\mathsf{P}}_s^a V$ are 1-Lipschitz, thus

$$\sigma_{\hat{\mathcal{P}}_s^a}(V^{s,u}) - \sigma_{\hat{\mathcal{P}}_s^a}(V^{s,u_i}) \leq \|V^{s,u} - V^{s,u_i}\|, \quad (119)$$

$$\mathsf{P}_s^a V^{s,u_i} - \mathsf{P}_s^a V^{s,u} \leq \|V^{s,u} - V^{s,u_i}\|. \quad (120)$$

Now from Equation (100), it follows that

$$\sigma_{\hat{\mathcal{P}}_s^a}(V^{s,u}) - \sigma_{\hat{\mathcal{P}}_s^a}(V^{s,u_i}) \leq \frac{1}{N(1-\gamma)}, \quad (121)$$

$$\mathsf{P}_s^a V^{s,u_i} - \mathsf{P}_s^a V^{s,u} \leq \frac{1}{N(1-\gamma)}. \quad (122)$$

Hence combine with Equation (118), and we have that

$$
\begin{aligned}
&\sigma_{\hat{\mathcal{P}}_s^a}(V_{\hat{\mathcal{P}}}^{\pi_r}) - \mathsf{P}_s^a V_{\hat{\mathcal{P}}}^{\pi_r} \\
&\leq 2\sqrt{\frac{48 \log \frac{4SAN}{(1-\gamma)\delta}\epsilon_1}{(1-\gamma)N(s,a)}} + 2\epsilon_1 \sqrt{\frac{48 \log \frac{4SAN}{(1-\gamma)\delta}}{N(s,a)}} + \frac{48 \log \frac{4SAN}{(1-\gamma)\delta}}{(1-\gamma)N(s,a)} \\
&\quad + \sigma_{\hat{\mathcal{P}}_s^a}(V^{s,u}) - \sigma_{\hat{\mathcal{P}}_s^a}(V^{s,u_i}) + \mathsf{P}_s^a V^{s,u_i} - \mathsf{P}_s^a V^{s,u} \\
&\leq 2\sqrt{\frac{48 \log \frac{4SAN}{(1-\gamma)\delta}\epsilon_1}{(1-\gamma)N(s,a)}} + 2\epsilon_1 \sqrt{\frac{48 \log \frac{4SAN}{(1-\gamma)\delta}}{N(s,a)}} + \frac{48 \log \frac{4SAN}{(1-\gamma)\delta}}{(1-\gamma)N(s,a)} + \frac{2}{N(1-\gamma)} \\
&\leq 2\sqrt{\frac{48 \log \frac{4SAN}{(1-\gamma)\delta}\epsilon_1}{(1-\gamma)N(s,a)}} + 2\epsilon_1 \sqrt{\frac{48 \log \frac{4SAN}{(1-\gamma)\delta}}{N(s,a)}} + \frac{96 \log \frac{4SAN}{(1-\gamma)\delta}}{(1-\gamma)N(s,a)}, \quad (123)
\end{aligned}
$$

which completes the proof. $\qquad \square$