# OpenReview forum: "Achieving Minimax Optimal Sample Complexity of Offline Reinforcement Learning: A DRO-Based Approach"
_ICLR.cc/2024/Conference — ICLR 2024 Conference Withdrawn Submission_

### Official Review · Reviewer_MQDJ · 2023-10-28

**Soundness:** 2 fair
**Presentation:** 3 good
**Contribution:** 2 fair
**Rating:** 3
**Confidence:** 4

**Summary:**

This paper proposes a distributionally robust optimization framework to solve offline tabular reinforcement learning problems with partial coverage. The paper designs two instances of uncertainty set, the first instance with total variation and Hoeffding inequality, and the improved instance with chi-square divergence and Bernstein inequality, which matches the minimax lower bound.

**Strengths:**

The paper brings forth a new perspective on addressing the challenges of offline reinforcement learning by drawing connections between pessimism and distributionally robustness. This conceptual link is an intriguing contribution and has the potential to serve as a catalyst for further exploration in this direction.

**Weaknesses:**

1. While the proposed algorithm is undoubtedly insightful, it faces limitations in terms of novelty, primarily stemming from its application within the domain of tabular reinforcement learning. Consequently, the theoretical results and computational advantages associated with this algorithm do not appear to significantly outperform existing pessimism-based methods.

2. Some aspects of the explanations provided in the paper may benefit from further rigor and clarity, as indicated by the questions that arise during the reading process.

**Questions:**

1. Could the authors provide a more explicit connection between the insight presented below Equation (7) – namely, "However, $\Delta_2=\min_{q\in \hat{\mathcal{P}}} E_q[\sum\gamma^t r_t ] - E_p[\sum \gamma^t r_t ] $" – and the improvements achieved through the Bernstein-style results? While I acknowledge that different distributions may indeed have the same expectations of reward, the paper appears to underutilize this insight in the subsequent analysis without specifying further structure in the reward. As the paper characters the worst-case suboptimality, any possible gap from two expected return from any two neighbor distributions should be included into considerations. It would be beneficial to explore how this observation can be further leveraged in the context of worst-case suboptimality characterization in a more rigorous (mathematical) manner.

2. I would appreciate a more comprehensive explanation regarding the advantages of employing both the chi-square divergence and the Bernstein-style uncertainty set. The paper suggests that the chi-square divergence, when compared to the total variation divergence with the same radius, is less conservative and can save on sample size (N^{-0.5}). However, it remains somewhat unclear how the use of the Bernstein-style radius and the Hoeffding-style radius contributes to saving (1-\gamma)^{-1}, which is the most salient improvement. Establishing a clearer connection between these elements and their implications on the chi-square divergence or the Bernstein inequality would enhance the understanding of their benefits.

---

### Official Review · Reviewer_VTaD · 2023-10-28

**Soundness:** 3 good
**Presentation:** 2 fair
**Contribution:** 2 fair
**Rating:** 5
**Confidence:** 3

**Summary:**

This paper studies the problem of offline reinforcement learning through the lens of distributional robust optimization. It proposes two methods, one based on Hoeffding inequality and one based on Bernstein inequality, and achieved bounds of $O(\epsilon^{-2}(1-\gamma)^{-4})$ and $O(\epsilon^{-2}(1-\gamma)^{-3})$ respectively.

**Strengths:**

The paper is very clearly written, and the analysis is complemented by extensive experiments to validate the claims empirically. The results for using Bernstein inequality with a chi-square uncertainty set are interesting, and they match with the lower bound presented by Yang et al. 2022. I believe this result is also new in robust reinforcement learning.

Yang, Wenhao, Liangyu Zhang, and Zhihua Zhang. "Toward theoretical understandings of robust Markov decision processes: Sample complexity and asymptotics." The Annals of Statistics 50.6 (2022): 3223-3248.

**Weaknesses:**

It is unclear to me whether the LCB approach and the DRO approach are equivalent. I am not sure if one can show that taking the worst-case transition from the uncertainty set with the designed bonus is equivalent to adding the pessimistic bonus to the value function. Given that the bounds achieved by the LCB approaches are tight and this paper claims to recover the bounds from LCB approaches, I believe it is important to understand whether the two methods are really different. Please see the questions for detailed elaboration on this.

**Questions:**

To my understanding, the LCB approach essentially draws an uncertainty set around $\hat{P}$ with the radius of the bonus function $b$. This bonus function should ensure that it provides an upper bound of $(\hat{P} - P)^\top V$, and thus the estimated value function $\hat{V}$ is a pessimistic estimate. This is very similar to the DRO approach in the sense that the uncertainty set from the DRO takes the role of the uncertainty set from the LCB approach. Indeed, if we take a look at the effect of the DRO uncertainty set in the analysis, it exactly ensures the pessimistic $V_{\hat{r}, \mathrm{P}}^\pi(s) \geq V_{\hat{r}, \hat{\mathcal{P}}}^\pi(s)=V_{\hat{\mathcal{P}}}^\pi(s)$ and gives rise to the following inequalities for the other terms. \begin{align*}
\left|b^*\left(V_{\hat{\mathcal{P}}}^{\pi_r}\right)(s)\right| =\gamma\left|\mathrm{P} _s^{\pi^*(s)} V _{\hat{\mathcal{P}}}^{\pi _r}-\sigma _{\hat{\mathcal{P}} _s^{\pi^*(s)}}\left(V _{\hat{\mathcal{P}}}^{\pi _r}\right)\right|
\leq\left\|\mathrm{P} _s^{\pi^*(s)}-\mathrm{Q} _s^{\pi^*(s)}\right\|_1\left\|V _{\hat{\mathcal{P}}}^{\pi_r}\right\| _{\infty}
\leq \frac{1}{1-\gamma} \sqrt{\frac{2 \log \frac{S A}{\delta}}{N\left(s, \pi^*(s)\right)}} .
\end{align*}
I believe the case is similar in the Bernstein-style uncertainty set, where the Chi-squared divergence gives you the variance term to better leverage the Bernstein inequality.  I am very curious whether one can set $\sqrt{\frac{2 \log \frac{S A}{\delta}}{N\left(s, \pi^*(s)\right)}}$ as the LCB bonus term and achieve the same result, and similarly for the case of Bernstein-style uncertainty set. The constant terms might be a bit different, but the rest should look the same.

I wonder if the authors can further elaborate on the details of the difference between the LCB and DRO approaches, and show that one cannot be reduced to the other?  I am happy to raise my rating if this can be clarified.

---

### Official Review · Reviewer_ZsAy · 2023-10-31

**Soundness:** 3 good
**Presentation:** 3 good
**Contribution:** 2 fair
**Rating:** 5
**Confidence:** 3

**Summary:**

This paper is dealing with the offline reinforcement learning problem with finite states and finite actions. It shows that the distributionally robust optimization based approach could achieve the optimal minimax sample complexity by directly modeling the uncertainty in the transition kernels. The optimal minimax sample complexity is achieved by constructing the Bernstein-style uncertainty set and using the Chi-square divergence as the distance measure. Some examples are given to show that the proposed algorithm has very close performance as previous minimax optimal method for different sample sizes.

**Strengths:**

1. The paper shows that the distributionally robust optimization based approach could also achieve minimax optimal sample complexity if the uncertainty set and distance measurement are chosen wisely.
2. The experiment examples show very close performance for different sample sizes when compared with the Lower Confidence Bound (LCB) approach.
3. The paper is well-written and easy to follow.

**Weaknesses:**

1. The chosen uncertainty set has the same flavor as the previously minimax optimal method based on the Lower Confidence Bound. Given the known optimal performance from the LCB method and previous analysis about the DRO approach in offline RL, this paper's result seems not hard to derive and is well expected.
2. The main contribution of this paper seems to be like making a good choice of the uncertainty set and the associated distance measure. Since the previous LCB method has shown the Bernstein-style uncertainty set could lead to optimal performance, and the value (or action-value) function is linear in both reward and transition kernel, it feels straightforward to construct this type of uncertainty set to get the optimal result.

**Questions:**

Could the author/s make some comments on my review about weakness?

---

### Official Review · Reviewer_tZBG · 2023-10-31

**Soundness:** 2 fair
**Presentation:** 4 excellent
**Contribution:** 3 good
**Rating:** 3
**Confidence:** 4

**Summary:**

Distributionally robust reinforcement learning (DR-RL) has attracted increasing attention. The underlying RMDP framework aims to find the best policy under the worst model in a pre-specified uncertainty set. This is inherently pessimistic. On the other hand, distribution shift issue in offline RL is often tackled by introducing pessimism and preventing us from overestimating the values. The paper connects these two by using distributionally robust optimization to solve non-robust offline RL problem. Finite sample complexity guarantee is provided, along with simulations on tabular problems.

**Strengths:**

1. The 2021 paper [1] proposed a new kind of pessimism based on model for offline RL, instead of reward-based pessimism (e.g. LCB) in prior offline RL works. [1] did not provide a tractable algorithm but hinted that DRO framework could be brought in as a solution for this approach. Now I think the main contribution of this paper is completing the connection between DRO (DR-RL) and offline RL. In particular, it materializes the abstract algorithm presented in [1]. It also admits a single-policy concentrability (partial coverage assumption), on par with [1] and other state-of-the-art offline RL algorithms.
2. The extensive analysis going from Hoeffding to Bernstein is insightful, and it is a meaningful contribution.
3. The implication of this work can be profound in that this connection enables one to bring new techniques developed in DRO literature to solving offline RL problems, including tractable optimization algorithms.

[1] M. Uehara and W. Sun. Pessimistic model-based offline reinforcement learning under partial coverage. arXiv preprint arXiv:2107.06226, 2021.

**Weaknesses:**

I have some concerns regarding the proof and the correctness of some results. I think the authors made a critical but fixable mistake in the Hoeffding part. See questions below.

**Questions:**

1. We know that $d_\mathrm{TV}(p,\hat{p}) = (1/2)\lVert p-\hat{p}\rVert_1$ And it's well-known that there is no dimension-free concentration result for $l^1$ norm. The authors made a mistake when they cite the Hoeffding's inequality in Eq. 10. Intuitively, you are summing up deviations in each coordinate, and the right hand side of Eq. 10 is only the vanilla Hoeffding's inequality bound which is only about the concentration of the empirical mean. See [A] for a simple proof of the concentration of the total variation distance  between the true and learned (empirical) distribution. The consequence is that your Hoeffding sample complexity will now scale with $S^2$.

2. I observe that $\mu_\mathrm{min}$ potentially scales with $SA$ when the data generating distribution is a generative model. This could make the worst-case sample complexity in Eq. 9 at least $S^2A^2$ after burn-in cost. Am I missing anything?

In conclusion, I have concern about whether the sample complexity result presented in this paper is really minimax optimal. I look forward to discussing with the authors. Thank you.

[A} Canonne, C. L. (2020). A short note on learning discrete distributions. arXiv preprint arXiv:2002.11457.